# ANCER: ANISOTROPIC CERTIFICATION VIA SAMPLE-WISE VOLUME MAXIMIZATION

## ABSTRACT

Randomized smoothing has recently emerged as an effective tool that enables certification of deep neural network classifiers at scale. All prior art on randomized smoothing has focused on isotropic $\ell_p$ certification, which has the advantage of yielding certificates that can be easily compared among isotropic methods via $\ell_p$-norm radius. However, isotropic certification limits the region that can be certified around an input to *worst-case* adversaries, *i.e.* it cannot reason about other "close", potentially large, constant prediction safe regions. To alleviate this issue, (**i**) we theoretically extend the isotropic randomized smoothing $\ell_1$ and $\ell_2$ certificates to their generalized *anisotropic* counterparts following a simplified analysis. Moreover, (**ii**) we propose evaluation metrics allowing for the comparison of general certificates – a certificate is superior to another if it certifies a *superset* region – with the quantification of each certificate through the volume of the certified region. We introduce ANCER, a framework for obtaining anisotropic certificates for a given test set sample via volume maximization. We achieve it by generalizing memory-based certification of data-dependent classifiers. Our empirical results demonstrate that ANCER achieves state-of-the-art $\ell_1$ and $\ell_2$ certified accuracy on CIFAR-10 and ImageNet in the data-dependence setting, while certifying larger regions in terms of volume, highlighting the benefits of moving away from isotropic analysis.

## 1 INTRODUCTION

The well-studied fact that Deep Neural Networks (DNNs) are vulnerable to additive imperceptible noise perturbations has led to a growing interest in developing robust classifiers (Goodfellow et al., 2015; Szegedy et al., 2014). A recent promising approach to achieve state-of-the-art provable robustness (*i.e.* a theoretical bound on the output around every input) at the scale of ImageNet (Deng et al., 2009) is *randomized smoothing* (Lecuyer et al., 2019; Cohen et al., 2019). Given an input $x$ and a network $f$, randomized smoothing constructs $g(x) = \mathbb{E}_{\epsilon \sim \mathcal{D}}[f(x + \epsilon)]$ such that $g(x) = g(x + \delta) \ \forall \delta \in \mathcal{R}$, where the certification region $\mathcal{R}$ is characterized by $x$, $f$, and the smoothing distribution $\mathcal{D}$. For instance, Cohen et al. (2019) showed that if $\mathcal{D} = \mathcal{N}(0, \sigma^2 I)$, then $\mathcal{R}$ is an $\ell_2$-ball whose radius is determined by $x$, $f$ and $\sigma$. Since then, there has been significant progress towards the design of $\mathcal{D}$ leading to the largest $\mathcal{R}$ for all inputs $x$. The interplay between $\mathcal{R}$ characterized by $\ell_1$, $\ell_2$ and $\ell_\infty$-balls, and a notion of optimal distribution $\mathcal{D}$ has been previously studied Yang et al. (2020).

Despite this progress, current randomized smoothing approaches provide certification regions that are *isotropic* in nature, limiting their capacity to certifying smaller and *worst-case* regions. We provide an intuitive example of this behavior in Figure 1. The isotropic nature of $\mathcal{R}$ in prior art is due to the common assumption that the smoothing distribution $\mathcal{D}$ is identically distributed (Yang et al., 2020; Kumar et al., 2020; Levine & Feizi, 2021). Moreover, comparisons between various randomized smoothing approaches were limited to methods that produce the same $\ell_p$ certificate, with no clear metrics for comparing with other certificates. In this paper, we address both concerns and present new state-of-the-art certified accuracy results on both CIFAR-10 and ImageNet datasets.

Our contributions are threefold. (**i**) We provide a general and simpler analysis compared to prior art (Cohen et al., 2019; Yang et al., 2020) that paves the way for the certification of *anisotropic* regions characterized by any norm, holding prior art as special cases. We then specialize our result to regions that, for a positive definite $\mathbf{A}$, are ellipsoids, *i.e.* $\|\mathbf{A}\delta\|_2 \leq c$, and generalized cross-polytopes,

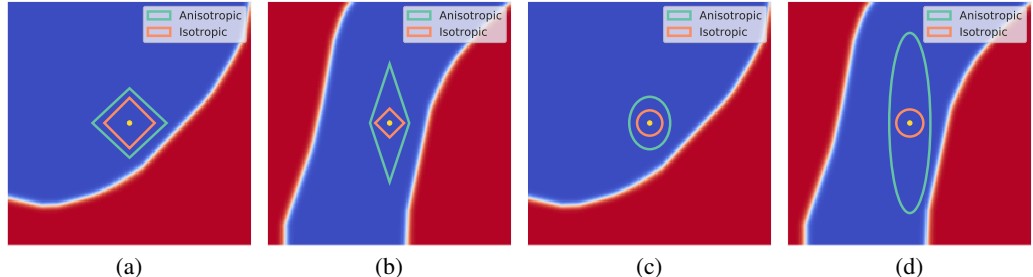

Figure 1: Illustration of the landscape of $f^y$ (blue corresponds to a higher confidence in $y$, the true label) for a region around an input in a toy, 2-dimensional radially separable dataset. For two dataset examples, in (a) and (b) we show the boundaries of the optimal $\ell_1$ isotropic and anisotropic certificates, while (c) and (d) are the boundaries of the optimal $\ell_2$ isotropic and anisotropic certificates. A thorough discussion of this figure is presented in Section 3.

*i.e.* $\|\mathbf{A}\delta\|_1 \leq c$, generalizing both $\ell_2$ (Cohen et al., 2019) and $\ell_1$ (Lecuyer et al., 2019; Yang et al., 2020) certification (Section 4). **(ii)** We introduce a new evaluation framework to compare methods that certify general (isotropic or anisotropic) regions. We compare two general certificates by defining that a method certifying $\mathcal{R}_1$ is superior to another certifying $\mathcal{R}_2$, if $\mathcal{R}_1$ is a strict superset to $\mathcal{R}_2$. Further, we define a standalone quantitative metric as the volume of the certified region, and specialize it for the cases of ellipsoids and generalized cross-polytopes (Section 5). **(iii)** We propose ANCER, an anisotropic certification method that performs sample-wise (*i.e.* per sample in the test set) region volume maximization (Section 6), generalizing the data-dependent, memory-based solution from Alfarra et al. (2020). Through experiments on CIFAR-10 (Krizhevsky, 2009) and ImageNet (Deng et al., 2009), we show that restricting ANCER's certification region to $\ell_1$ and $\ell_2$-balls outperforms state-of-the-art $\ell_1$ and $\ell_2$ results from previous works (Yang et al., 2020; Alfarra et al., 2020). Further, we show that the volume of the certified regions are significantly larger than all existing methods, thus setting a new state-of-the-art in certified accuracy. We highlight that while we effectively achieve state-of-the-art performance, it comes at a high cost given the data-dependency requirements. A discussion of the limitations of the solution is presented in Section 6.

**Notation.** We consider a base classifier $f : \mathbb{R}^n \to \mathcal{P}(K)$, where $\mathcal{P}(K)$ is a probability simplex over $K$ classes, *i.e.* $f^i \geq 0$ and $\mathbf{1}^\top f = 1$, for $i \in \{1, \dots, K\}$. Further, we use $(x, y)$ to be a sample input $x$ and its corresponding true label $y$ drawn from a test set $\mathcal{D}_t$, and $f^y$ to be the output of $f$ at the correct class. We use $\ell_p$ to be the typically defined $\|\cdot\|_p$ norm ($p \geq 1$), and $\ell_p^{\mathbf{A}}$ or $\|\cdot\|_{\mathbf{A},p}$ for $p = \{1, 2\}$ to be a composite norm defined with respect to a positive definite matrix $\mathbf{A}$ as $\|\mathbf{A}^{-1/p}v\|_p$.

## 2 RELATED WORK

**Verified Defenses.** Since the discovery that DNNs are vulnerable against input perturbations (Goodfellow et al., 2015; Szegedy et al., 2014), a range of methods have been proposed to build classifiers that are verifiably robust (Huang et al., 2017; Gowal et al., 2019; Bunel et al., 2018; Salman et al., 2019b). Despite this progress, these methods do not yet scale to the networks the community is interested in certifying (Tjeng et al., 2019; Weng et al., 2018).

**Randomized Smoothing.** The first works on randomized smoothing used Laplacian (Lecuyer et al., 2019; Li et al., 2019) and Gaussian Cohen et al. (2019) distributions to obtain $\ell_1$ and $\ell_2$-ball certificates, respectively. Several subsequent works improved the performance of smooth classifiers by training the base classifier using adversarial augmentation (Salman et al., 2019a), regularization (Zhai et al., 2019), or general adjustments to training routines (Jeong & Shin, 2020). Recent work derived $\ell_p$-norm certificates for other isotropic smoothing distributions (Yang et al., 2020; Levine & Feizi, 2020; Zhang et al., 2019). Concurrently, Dvijotham et al. (2020) developed a framework to handle arbitrary smoothing measures in any $\ell_p$-norm; however, the certification process requires significant hyperparameter tuning. Similarly, Mohapatra et al. (2020) introduces larger certificates that require higher-order information, yet do not provide a closed-form solution. This was followed by a complementary data-dependent smoothing approach, where the parameters of the smoothing distribution were optimized per test set *sample* to maximize the certified radius at an individual

input (Alfarra et al., 2020). All prior works considered smoothing with *isotropic* distributions and hence certified isotropic $\ell_p$-ball regions. In this paper, we extend randomized smoothing to certify *anisotropic* regions, by pairing it with a generalization of the data-dependent framework (Alfarra et al., 2020) to maximize the certified region at each input point.

## 3   MOTIVATING ANISOTROPIC CERTIFICATES

Certification approaches aim to find the *safe* region $\mathcal{R}$, where $\arg\max_i f^i(x) = \arg\max_i f^i(x + \delta) \; \forall \delta \in \mathcal{R}$. Recent randomized smoothing techniques perform this certification by explicitly optimizing the isotropic $\ell_p$ certified region around each input (Alfarra et al., 2020), obtaining state-of-the-art performance as a result. Despite this $\ell_p$ optimality, we note that any $\ell_p$-norm certificate is *worst-case* from the perspective of that norm, as it avoids adversary regions by limiting its certificate to the $\ell_p$-closest adversary. This means that it can only enjoy a radius that is at most equal to the distance to the closest decision boundary. However, decision boundaries of general classifiers are complex, non-linear, and non-radially distributed with respect to a generic input sample (Karimi et al., 2019). This is evidenced by the fact that, within a reasonably small $\ell_p$-ball around an input, there are often only a small set of adversary directions (Tramèr et al., 2017; 2018) (*e.g.* see the decision boundaries in Figure 1). As such, while $\ell_p$-norm certificates are useful to reason about worst-case performance and are simple to obtain given previous works (Cohen et al., 2019; Yang et al., 2020; Lee et al., 2019), they are otherwise uninformative in terms of the shape of decision boundaries, *i.e.* which regions around the input are safe.

To visualize these concepts, we illustrate the decision boundaries of a base classifier $f$ trained on a toy 2-dimensional, radially separable (with respect to the origin) binary classification dataset, and consider two different input test samples (see Figure 1). We compare the *optimal* isotropic and anisotropic certified regions of different shapes at these points. In Figures 1a and 1b, we compare an isotropic cross-polytope (of the form $\|\delta\|_1 \leq r$) with an anisotropic generalized cross-polytope (of the form $\|\mathbf{A}\delta\|_1 \leq r$), while in Figures 1c and 1d we compare an isotropic $\ell_2$ ball (of the form $\|\delta\|_2 \leq r$) with an anisotropic ellipsoid (of the form $\|\mathbf{A}\delta\|_2 \leq r$). Notice that in Figures 1a and 1c, due to the curvature of the classification boundary (shown in white), the optimal certification region is isotropic in nature, which is evidenced by the similarities of the optimal isotropic and anisotropic certificates. On the other hand, in Figures 1b and 1d, the location of the decision boundary allows for the anisotropic certified regions to be considerably larger than their isotropic counterparts, as they are not as constrained by the closest decision boundary, *i.e.* the *worst-case* performance. We note that these differences are further highlighted in higher dimensions, and we study them for a single CIFAR-10 test set sample in Appendix A.1. Further, we also showcase how anisotropic certification allows for further insights into constant prediction (safe) directions in Appendix A.2.

## 4   ANISOTROPIC CERTIFICATION

One of the main obstacles in enabling anisotropic certification is the complexity of the analysis required. To alleviate this, we follow a Lipschitz argument first observed by Salman et al. (2019a) and Jordan & Dimakis (2020) and propose a simple and general certification analysis. We start with the following two observations. All proofs are in Appendix B.

**Proposition 1.** *Consider a differentiable function $g : \mathbb{R}^n \to \mathbb{R}$. If $\sup_x \|\nabla g(x)\|_* \leq L$ where $\|\cdot\|_*$ has a dual norm $\|z\| = \max_x z^\top x \;\; s.t. \; \|x\|_* \leq 1$, then $g$ is $L$-Lipschitz under norm $\|\cdot\|_*$, that is $|g(x) - g(y)| \leq L\|x - y\|$.*

Given the previous proposition, we formalize $\|\cdot\|$ certification as follows:

**Theorem 1.** *Let $g : \mathbb{R}^n \to \mathbb{R}^K$, $g^i$ be $L$-Lipschitz continuous under norm $\|\cdot\|_* \; \forall i \in \{1, \ldots, K\}$, and $c_A = \arg\max_i g^i(x)$. Then, we have $\arg\max_i g^i(x + \delta) = c_A$ for all $\delta$ satisfying:*

$$\|\delta\| \leq \frac{1}{2L}\left(g^{c_A}(x) - \max_c g^{c \neq c_A}(x)\right).$$

Theorem 1 provides an $\|\cdot\|$ norm robustness certificate for any $L$-Lipschitz classifier $g$ under $\|\cdot\|_*$. The certificate is only informative when one can attain a tight *non-trivial* estimate of $L$, ideally $\sup_x \|\nabla g(x)\|_*$, which is generally difficult when $g$ is an arbitrary neural network.

**Framework Recipe.** In light of Theorem 1, randomized smoothing can be viewed **differently** as an instance of Theorem 1 with the favorable property that the constructed smooth classifier $g$ enjoys an analytical form for $L = \sup_x \|\nabla g(x)\|_*$ by design. As such, to obtain an informative $\|\cdot\|$ certificate, one must, for an arbitrary choice of a smoothing distribution, compute the analytic Lipschitz constant $L$ under $\|\cdot\|_*$ for the smooth $g$. While there can exist a notion of "optimal" smoothing distribution for a given choice of $\|\cdot\|$ certificate, as in part addressed earlier for the isotropic $\ell_1$, $\ell_2$ and $\ell_\infty$ certificates (Yang et al., 2020), this is not the focus of this paper. The choice of the smoothing distribution in later sections is inspired by previous work for the purpose of granting anisotropic certificates. This recipe complements randomized smoothing works based on Neyman-Pearson's lemma (Cohen et al., 2019) or the Level-Set and Differential Method (Yang et al., 2020).

We will deploy this framework recipe to show two specializations for anisotropic certification, namely ellipsoids (Section 4.1) and generalized cross-polytopes (Section 4.2).[1]

## 4.1 CERTIFYING ELLIPSOIDS

In this section, we consider the certification under $\ell_2^\Sigma$ norm, or $\|\delta\|_{\Sigma,2} = \sqrt{\delta^\top \Sigma^{-1} \delta}$, that has a dual norm $\|\delta\|_{\Sigma^{-1},2}$. Note that both $\|\delta\|_{\Sigma,2} \le r$ and $\|\delta\|_{\Sigma^{-1},2} \le r$ define an ellipsoid. Despite that the following results hold for any positive definite $\Sigma$, we assume for efficiency reasons that $\Sigma$ is diagonal throughout. First, we consider the anisotropic Gaussian smoothing distribution $\mathcal{N}(0,\Sigma)$ with the smooth classifier defined as $g_\Sigma(x) = \mathbb{E}_{\epsilon \sim \mathcal{N}(0,\Sigma)}[f(x+\epsilon)]$. Considering the classifier $\Phi^{-1}(g_\Sigma(x))$, where $\Phi$ is the standard Gaussian CDF, and following Theorem 1 to grant an $\ell_2^\Sigma$ certificate for $\Phi^{-1}(g_\Sigma(x))$, we derive the Lipschitz constant $L$ under $\|\cdot\|_{\Sigma^{-1},2}$, in the following proposition.

**Proposition 2.** $\Phi^{-1}(g_\Sigma(x))$ *is 1-Lipschitz (i.e. $L = 1$) under the $\|\cdot\|_{\Sigma^{-1},2}$ norm.*

Since $\Phi^{-1}$ is a strictly increasing function, by combining Proposition 2 with Theorem 1, we have:

**Corollary 1.** *Let $c_A = \arg\max_i g_\Sigma^i(x)$ , then $\arg\max_i g_\Sigma^i(x+\delta) = c_A$ for all $\delta$ satisfying:*

$$\|\delta\|_{\Sigma,2} \le \frac{1}{2}\left(\Phi^{-1}\left(g_\Sigma^{c_A}(x)\right) - \Phi^{-1}\left(\max_c g_\Sigma^{c \ne c_A}(x)\right)\right).$$

Corollary 1 holds the $\ell_2$ certification from Zhai et al. (2019) as a special case for when $\Sigma = \sigma^2 I$.[2]

## 4.2 CERTIFYING GENERALIZED CROSS-POLYTOPES

Here we consider certification under the $\ell_1^\Lambda$ norm defining a generalized cross-polytope, *i.e.* the set $\{\delta : \|\delta\|_{\Lambda,1} = \|\Lambda^{-1}\delta\|_1 \le r\}$, as opposed to the $\ell_1$-bounded set that defines a cross-polytope, *i.e.* $\{\delta : \|\delta\|_1 \le r\}$. As with the ellipsoid case and despite that the following results hold for any positive definite $\Lambda$, for the sake of efficiency, we assume $\Lambda$ to be diagonal throughout. For generalized cross-polytope certification, we consider an anisotropic Uniform smoothing distribution $\mathcal{U}$, which defines the smooth classifier $g_\Lambda(x) = \mathbb{E}_{\epsilon \sim \mathcal{U}[-1,1]^n}[f(x+\Lambda\epsilon)]$. Following Theorem 1 and to certify under the $\ell_1^\Lambda$ norm, we compute the Lipschitz constant of $g_\Lambda$ under the $\|\Lambda x\|_\infty$ norm, which is the dual norm of $\|\cdot\|_{\Lambda,1}$ (see Appendix B), in the next proposition.

**Proposition 3.** *The classifier $g_\Lambda$ is $1/2$-Lipschitz (i.e. $L = 1/2$) under the $\|\Lambda x\|_\infty$ norm.*

Similar to Corollary 1, by combining Proposition 3 with Theorem 1, we have that:

**Corollary 2.** *Let $c_A = \arg\max_i g_\Lambda^i(x)$ , then $\arg\max_i g_\Lambda^i(x+\delta) = c_A$ for all $\delta$ satisfying:*

$$\|\delta\|_{\Lambda,1} = \|\Lambda^{-1}\delta\|_1 \le \left(g_\Lambda^{c_A}(x) - \max_c g_\Lambda^{c \ne c_A}(x)\right).$$

Corollary 2 holds the $\ell_1$ certification from Yang et al. (2020) as a special case for when $\Lambda = \lambda I$.

---

[1]Our analysis also grants a certificate for a mixture of Gaussians smoothing distribution (see Appendix B.1).
[2]A similar result was derived in the appendix of Fischer et al. (2020); Li et al. (2020) with a more involved analysis by extending Neyman-Pearson's lemma.

## 5 EVALUATING ANISOTROPIC CERTIFICATES

With the anisotropic certification framework presented in the previous section, the question arises: "Given two general (isotropic or anisotropic) certification regions $\mathcal{R}_1$ and $\mathcal{R}_2$, how can one effectively compare them?". We propose the following definition to address this issue.

**Definition 1.** *For a given input point $x$, consider the two certification regions $\mathcal{R}_1$ and $\mathcal{R}_2$ obtained for two classifiers $f_1$ and $f_2$, i.e. $\mathcal{A}_1 = \{\delta : \arg\max_c f_1^c(x) = \arg\max_c f_1^c(x+\delta), \forall \delta \in \mathcal{R}_1\}$ and $\mathcal{A}_2 = \{\delta : \arg\max_c f_2^c(x) = \arg\max_c f_2^c(x+\delta), \forall \delta \in \mathcal{R}_2\}$ where $\arg\max_c f_1^c(x) = \arg\max_c f_2^c(x)$. We say $\mathcal{A}_1$ is a "superior certificate" to $\mathcal{A}_2$ (i.e. $\mathcal{A}_1 \succ \mathcal{A}_2$), if and only if, $\mathcal{A}_1 \supset \mathcal{A}_2$.*

This definition is a natural extension from the radius-based comparison of $\ell_p$-ball certificates, providing a basis for evaluating anisotropic certification. To compare an anisotropic to an isotropic region of certification, it is not immediately clear how to **(i)** check that an anisotropic region is a superset to the isotropic region, and **(ii)** if it were a superset, how to quantify the improvement of the anisotropic region over the isotropic counterpart. In Sections 5.1 and 5.2, we tackle these issues for the particular cases of ellipsoid and generalized cross-polytope certificates.

### 5.1 EVALUATING ELLIPSOID CERTIFICATES

**Comparing $\ell_2-$Balls to $\ell_2^\Sigma-$Ellipsoids (Specialization of Definition 1).** Recall that if $\Sigma = \sigma^2 I$, our ellipsoid certification in Corollary 1 recovers as a special case the isotropic $\ell_2$-ball certification of Cohen et al. (2019); Salman et al. (2019a); Zhai et al. (2019). Consider the certified regions $\mathcal{R}_1 = \{\delta : \|\delta\|_2 \le \tilde{\sigma} r_1\}$ and $\mathcal{R}_2 = \{\delta : \|\delta\|_{\Sigma,2} = \sqrt{\delta^\top \Sigma^{-1}\delta} \le r_2\}$ for given $r_1, r_2 > 0$. Since we take $\Sigma = \mathrm{diag}(\{\sigma_i^2\}_{i=1}^n)$, the maximum enclosed $\ell_2$-ball for the ellipsoid $\mathcal{R}_2$ is given by the set $\mathcal{R}_3 = \{\delta : \|\delta\|_2 \le \min_i \sigma_i r_2\}$, and thus $\mathcal{R}_2 \supseteq \mathcal{R}_3$. Therefore, it suffices that $\mathcal{R}_3 \supseteq \mathcal{R}_1$ (*i.e.* $\min_i \sigma_i r_2 \ge \tilde{\sigma} r_1$), to say that $\mathcal{R}_2$ is a superior certificate to the isotropic $\mathcal{R}_1$ as per Definition 1.

**Quantifying $\ell_2^\Sigma$ Certificates.** The aforementioned specialization is only concerned with whether our ellipsoid certified region $\mathcal{R}_2$ is "superior" to the isotropic $\ell_2$-ball without quantifying it. A natural solution is to directly compare the volumes of the certified regions. Since the volume of an ellipsoid given by $\mathcal{R}_2$ is $\mathcal{V}(\mathcal{R}_2) = r_2^n \sqrt{\pi^n}/\Gamma(n/2+1) \prod_{i=1}^n \sigma_i$ (Kendall, 2004), we directly compare the *proxy radius* $\tilde{R}$ defined for $\mathcal{R}_2$ as $\tilde{R} = r_2 \sqrt[n]{\prod_i^n \sigma_i}$, since larger $\tilde{R}$ correspond to certified regions with larger volumes. Note that $\tilde{R}$, which is the $n^{\text{th}}$ root of the volume up to a constant factor, can be seen as a generalization to the certified radius in the case when $\sigma_i = \sigma \; \forall i$.

### 5.2 EVALUATING GENERALIZED CROSS-POLYTOPE CERTIFICATES

**Comparing $\ell_1-$Balls to $\ell_1^\Lambda-$Generalized Cross-Polytopes (Specialization of Definition 1).** Consider the certificates $\mathcal{S}_1 = \{\delta : \|\delta\|_1 \le \tilde{\lambda} r_1\}$, $\mathcal{S}_2 = \{\delta : \|\delta\|_{\Lambda,1} = \|\Lambda^{-1}\delta\|_1 \le r_2\}$, and $\mathcal{S}_3 = \{\delta : \|\delta\|_1 \le \min_i \lambda_i r_2\}$, where we take $\Lambda = \mathrm{diag}(\{\lambda_i\}_{i=1}^n)$. Note that since $\mathcal{S}_2 \supseteq \mathcal{S}_3$, then as per Definition 1, it suffices that $\mathcal{S}_3 \supseteq \mathcal{S}_1$ (*i.e.* $\min_i \lambda_i r_2 \ge \tilde{\lambda} r_1$) to say that the anisotropic generalized cross-polytope $\mathcal{S}_2$ is superior to the isotropic $\ell_1$-ball $\mathcal{S}_1$.

**Quantifying $\ell_1^\Lambda$ Certificates.** Following the approach proposed in the $\ell_2^\Sigma$ case, we quantitatively compare the generalized cross-polytope certification of Corollary 2 to the $\ell_1$ certificate through the volumes of the two regions. We first present the volume of the generalized cross-polytope.

**Proposition 4.** $\mathcal{V}\left(\{\delta : \|\Lambda^{-1}\delta\|_1 \le r\}\right) = \frac{(2r)^n}{n!}\prod_i \lambda_i.$

Following this definition, we define the *proxy radius* for $\mathcal{S}_2$ in this case to be $\tilde{R} = r_2 \sqrt[n]{\prod_{i=1}^n \lambda_i}$. As with the $\ell_2$ case, larger $\tilde{R}$ correspond certified regions with larger volumes. As in the ellipsoid case, $\tilde{R}$ can be seen as a generalization to the certified radius when $\lambda_i = \lambda \; \forall i$.

# 6 ANCER: SAMPLE-WISE VOLUME MAXIMIZATION FOR ANISOTROPIC CERTIFICATION

Given the results from the previous sections, we are now equipped to certify anisotropic regions, in particular ellipsoids and generalized cross-polytopes. As mentioned in Section 4, these regions are generally defined as $\mathcal{R} = \{\delta : \|\delta\|_{\Theta,p} \leq r^p\}$ for a given parameter of the smoothing distribution $\Theta = \mathrm{diag}\left(\{\theta_i\}_{i=1}^n\right)$, an $\ell_p$-norm ($p \in \{1,2\}$), and a *gap* value of $r^p \in \mathbb{R}^+$. At this point, one could simply take an anisotropic distribution with arbitrarily chosen parameters $\Theta$ and certify a trained network at any input point $x$, in the style of what was done in the previous randomized smoothing literature with isotropic distributions. However, the choice of $\Theta$ is more complex in the anisotropic case. A fixed choice of anisotropic $\Theta$ could severely underperform the isotropic case – take, for example, the anisotropic distribution of Figure 1d applied to the input of Figure 1c.

Instead of taking a fixed $\Theta$, we generalize the framework introduced in Alfarra et al. (2020), where parameters of the smoothing distribution are optimized per input test point (*i.e.* in a *sample-wise* fashion) so as to maximize the resulting certificate. The goal of the optimization in Alfarra et al. (2020) is, at a point $x$, to maximize the isotropic $\ell_2$ region described in Section 4.1 (*i.e.* $\{\delta : \|\delta\|_2 \leq \sigma^x r^p(x, \sigma^x))\}$), where $r^p$ is the gap and a function of $x$ and $\sigma^x \in \mathbb{R}^+$. In the isotropic $\ell_p$ case, this generalizes to maximizing the region $\{\delta : \|\delta\|_p \leq \theta^x r^p(x, \theta^x)\}$, which can be achieved by maximizing radius $\theta^x r^p(x, \theta^x)$ through $\theta^x \in \mathbb{R}^+$, obtaining $r^*_{\mathrm{iso}}$ (Alfarra et al., 2020).

For the general anisotropic case, we propose ANCER, whose objective is to maximize the volume of the certified region through the *proxy radius*, while satisfying the *superset* condition with respect to the maximum isotropic $\ell_2$ radius, $r^*_{\mathrm{iso}}$. In the case of the ellipsoids and generalized cross-polytopes as presented in Sections 5.1 and 5.2, respectively, ANCER's optimization problem can be written as:

$$\arg\max_{\Theta^x} \ r^p(x, \Theta^x) \sqrt[n]{\prod_i \theta_i^x} \qquad \text{s.t.} \quad \min_i \theta_i^x r^p(x, \Theta^x) \geq r^*_{\mathrm{iso}} \tag{1}$$

where $r^p(x, \Theta^x)$ is the gap value under the anisotropic smoothing distribution. We iteratively solve a relaxed version of Equation (1), with further details presented in Appendix C.

**Memory-based Anisotropic Certification.** While each of the classifiers induced by the parameter $\Theta^x$, i.e. $g_{\Theta^x}$, is robust by definition as presented in Section 4, the certification of the overall data-dependent classifier is not necessarily sound due to the optimization procedure for each $x$. This is a known issue in certifying data-dependent classifiers, and is addressed by Alfarra et al. (2020) through the use of a memory-based procedure. In Appendix D, we present an adapted version of this algorithm to ANCER. All subsequent results are obtained following this procedure.

**Limitations of ANCER.** Given ANCER uses a memorization procedure similar to the one presented in Alfarra et al. (2020), it incurs limitations on memory and runtime complexity. The main limitations of the memory-based certification are outlined in Appendix E of Alfarra et al. (2020). The anisotropic case increases on the complexity of the isotropic framework by the increased runtime of specific functions presented in Appendix D. Certification runtime comparisons are in Section 7.3.

Further, note that in memory-based data-dependent certification there is a single procedure for both certification and inference in contrast with the fixed $\sigma$ setting from Cohen et al. (2019). While the linear runtime dependency on memory size might appear daunting for the deployment of such a system, there are a few factors that could mitigate the cost. Firstly, in practice the models deployed get regularly updated in deployment, and the memory should be reset in those situations. Secondly, there are possible solutions which might attain sublinear runtime for the post-certification stage, such as the application of $k$-d trees to reduce the space of comparisons and speed-up the process. As such, we believe ANCER to be suited to applications in offline scenarios, where improved robustness is desired and inference time is not a critical issue.

A further limitation of the memorization procedure has to do with the impact of the order in which inputs are certified on the overall statistics obtained. Within a memory-based framework, certifying $x_2$ with $x_1$ in memory can be different from certifying $x_1$ with $x_2$ in memory if they intersect. In practice, given the low number of intersections observed with the original certified regions, this effect was almost negligible in the results presented in Section 7. For fairness of comparison with non-memory based methods, we report "worst-case" results for ANCER in which we abstain from deciding whenever an intersection of two certified regions occurs.

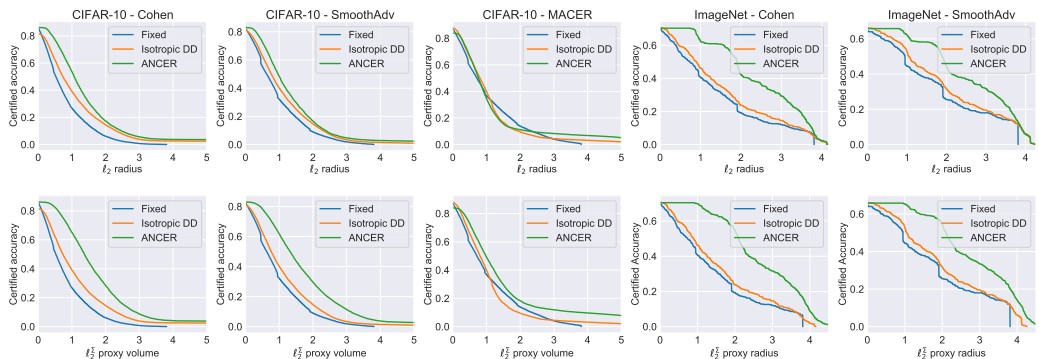

Figure 2: Distribution of top-1 certified accuracy as a function of $\ell_2$ radius (top) and $\ell_2^\Sigma$-norm proxy radius (bottom) obtained by different certification methods on CIFAR-10 and ImageNet.

## 7 EXPERIMENTS

We now study the empirical performance of ANCER to obtain $\ell_2^\Sigma$, $\ell_1^\Lambda$, $\ell_2$ and $\ell_1$ certificates on networks trained using randomized smoothing methods found in the literature. In this section, we show that ANCER is able to achieve **(i)** improved performance on those networks in terms of $\ell_2$ and $\ell_1$ certification when compared to certification baselines that smooth using a fixed isotropic $\sigma$ (Fixed $\sigma$) (Cohen et al., 2019; Yang et al., 2020; Salman et al., 2019a; Zhai et al., 2019) or a data-dependent and memory-based isotropic one (Isotropic DD) (Alfarra et al., 2020); and **(ii)** a significant improvement in terms of the $\ell_2^\Sigma$ and $\ell_1^\Lambda$-norm certified region obtained by the same methods – compared by computing the *proxy radius* of the certified regions – thus generally satisfying the conditions of a superior certificate proposed in Definition 1. Note that both data-dependent approaches (Isotropic DD and ANCER) use memory-based procedures. As such, the gains described in this section constitute a trade-off given the limitations of the method described in Section 6.

We follow an evaluation procedure as similar as possible to the ones described in Cohen et al. (2019); Yang et al. (2020); Salman et al. (2019a); Zhai et al. (2019) by using code and pre-trained networks whenever available and by performing experiments on CIFAR-10 (Krizhevsky, 2009) and ImageNet (Deng et al., 2009), certifying the entire CIFAR-10 test set and a subset of 500 examples from the ImageNet test set. For the implementation of ANCER, we solve Equation (1) with Adam for 100 iterations, where the certification gap $r^p(x, \Theta^x)$ is estimated at each iteration using 100 noise samples per test point (see Appendix C) and $\Theta^x$ in Equation (1) is initialized with the Isotropic DD solution from Alfarra et al. (2020). Further details of the setup can be found in Appendix E.

As in previous works, $\ell_p$ **certified accuracy** at radius $R$ is defined as the portion of the test set $\mathcal{D}_t$ for which the smooth classifier correctly classifies with an $\ell_p$ certification radius of at least $R$. In a similar fashion, we define the anisotropic $\ell_2^\Sigma/\ell_1^\Lambda$ certified accuracy at a proxy radius of $\tilde{R}$ (as defined in Section 5) to be the portion of $\mathcal{D}_t$ in which the smooth classifier classifies correctly with an $\ell_2^\Sigma/\ell_1^\Lambda$-norm certificate of an $n^{\text{th}}$ root volume of at least $\tilde{R}$. We also report **average certified radius** $(ACR)$ defined as $\mathbb{E}_{x,y\sim\mathcal{D}_t}[R_x \mathbb{1}(g(x) = y)]$ (Alfarra et al., 2020; Zhai et al., 2019) as well as **average certified proxy radius** $(AC\tilde{R})$ defined as $\mathbb{E}_{x,y\sim\mathcal{D}_t}[\tilde{R}_x \mathbb{1}(g(x) = y)]$, where $R_x$ and $\tilde{R}_x$ denote the radius and proxy radius at $x$ with a true label $y$ for a smooth classifier $g$. Recall that in the isotropic case, the proxy radius is, by definition, the same as the radius for a given $\ell_p$-norm. For each classifier, we ran experiments on the $\sigma$ values reported in the original work (with the exception of Yang et al. (2020), see Section 7.2). For the sake of brevity, we report in this section the top-1 certified accuracy plots, $ACR$ and $AC\tilde{R}$ per radius across $\sigma$, as in Salman et al. (2019a); Zhai et al. (2019); Alfarra et al. (2020). The performance of each method per $\sigma$ is presented in Appendix G.

### 7.1 ELLIPSOID CERTIFICATION ($\ell_2$ AND $\ell_2^\Sigma$-NORM CERTIFICATES)

We perform the comparison of $\ell_2$-ball vs. $\ell_2^\Sigma$-ellipsoid certificates via Gaussian smoothing using networks trained following the procedures defined in Cohen et al. (2019), Salman et al. (2019a), and Zhai et al. (2019). For each of these, we report results on ResNet18 trained using $\sigma \in \{0.12, 0.25, 0.5, 1.0\}$ for CIFAR-10, and ResNet50 using $\sigma \in \{0.25, 0.5, 1.0\}$ for ImageNet. For details of the training

Table 1: Comparison of top-1 certified accuracy at different $\ell_2$ radii, $\ell_2$ average certified radius ($ACR$) and $\ell_2^\Sigma$ average certified proxy radius ($A\tilde{CR}$) obtained by using the isotropic $\sigma$ used for training the networks (Fixed $\sigma$); the isotropic data-dependent (Isotropic DD) optimization scheme from Alfarra et al. (2020); and ANCER's data-dependent anisotropic optimization.

| **CIFAR-10** | Certification | Accuracy @ $\ell_2$ radius (%) | | | | | | | $\ell_2\ ACR$ | $\ell_2^\Sigma\ A\tilde{CR}$ |
|---|---|---|---|---|---|---|---|---|---|---|
| | | 0.0 | 0.25 | 0.5 | 1.0 | 1.5 | 2.0 | 2.5 | | |
| COHEN Cohen et al. (2019) | Fixed $\sigma$ | 86 | 71 | 51 | 27 | 14 | 6 | 2 | 0.722 | 0.722 |
| | Isotropic DD | 82 | 76 | 62 | 39 | 24 | 14 | 8 | 1.117 | 1.117 |
| | ANCER | 86 | 85 | 77 | 53 | 31 | 17 | 10 | **1.449** | **1.772** |
| SMOOTHADV Salman et al. (2019a) | Fixed $\sigma$ | 82 | 72 | 55 | 32 | 19 | 9 | 5 | 0.834 | 0.834 |
| | Isotropic DD | 82 | 75 | 63 | 40 | 25 | 15 | 7 | 1.011 | 1.011 |
| | ANCER | 83 | 81 | 73 | 48 | 30 | 17 | 8 | **1.224** | **1.573** |
| MACER Zhai et al. (2019) | Fixed $\sigma$ | 87 | 76 | 59 | 37 | 24 | 14 | 9 | 0.970 | 0.970 |
| | Isotropic DD | 88 | 80 | 66 | 40 | 17 | 9 | 6 | 1.007 | 1.007 |
| | ANCER | 84 | 80 | 67 | 34 | 15 | 11 | 9 | **1.136** | **1.481** |
| **ImageNet** | Certification | Accuracy @ $\ell_2$ radius (%) | | | | | | | $\ell_2\ ACR$ | $\ell_2^\Sigma\ A\tilde{CR}$ |
| | | 0.0 | 0.5 | 1.0 | 1.5 | 2.0 | 2.5 | 3.0 | | |
| COHEN Cohen et al. (2019) | Fixed $\sigma$ | 70 | 56 | 41 | 31 | 19 | 14 | 12 | 1.098 | 1.098 |
| | Isotropic DD | 71 | 59 | 46 | 36 | 24 | 19 | 15 | 1.234 | 1.234 |
| | ANCER | 70 | 70 | 62 | 61 | 42 | 36 | 29 | **1.810** | **1.981** |
| SMOOTHADV Salman et al. (2019a) | Fixed $\sigma$ | 65 | 59 | 44 | 38 | 26 | 20 | 18 | 1.287 | 1.287 |
| | Isotropic DD | 66 | 62 | 53 | 41 | 32 | 24 | 20 | 1.428 | 1.428 |
| | ANCER | 66 | 66 | 62 | 58 | 44 | 37 | 32 | **1.807** | **1.965** |

procedures, see Appendix E.1. Figure 2 plots top-1 certified accuracy as a function of the $\ell_2$ radius (top) and of the $\ell_2^\Sigma$-norm proxy radius (bottom) per trained network and dataset, while Table 1 presents an overview of the certified accuracy at various $\ell_2$ radii, as well as $\ell_2\ ACR$ and $\ell_2^\Sigma$-norm $A\tilde{CR}$. Recall that, following the considerations in Section 5.1, the $\ell_2$ certificate obtained through ANCER is the maximum enclosed isotropic $\ell_2$-ball in the $\ell_2^\Sigma$ ellipsoid.

First, we note that sample-wise certification (Isotropic DD and ANCER) achieves higher certified accuracy than fixed $\sigma$ across the board. This mirrors the findings in Alfarra et al. (2020), since certifying with a fixed $\sigma$ for all samples struggles with the robustness/accuracy trade-off first mentioned in Cohen et al. (2019), whereas the data-dependent solutions explicitly optimize $\sigma$ per sample to avoid it. More importantly, ANCER achieves new state-of-the-art $\ell_2$ certified accuracy at most radii in Table 1, *e.g.* at radius 0.5 ANCER brings certified accuracy to 77% (from 66%) and 70% (from 62%) on CIFAR-10 and ImageNet, respectively, yielding relative percentage improvements in $ACR$ between 13% and 47% when compared to Isotropic DD. While the results are significant, it might not be immediately clear why maximizing the volume of an ellipsoid with ANCER results in a larger maximum enclosed $\ell_2$-ball certificate in $\ell_2^\Sigma$ ellipsoid when compared to optimizing the $\ell_2$-ball with Isotropic DD. We explore this phenomenon in Appendix G.3.

As expected, ANCER substantially improves $\ell_2^\Sigma\ A\tilde{CR}$ compared to Isotropic DD in all cases – with relative improvements in $A\tilde{CR}$ between 38% and 63% over both datasets. The joint results, certification with $\ell_2$ and $\ell_2^\Sigma$, establish that ANCER certifies the $\ell_2$-ball region obtained by previous approaches, in addition to a much larger region captured by the $\ell_2^\Sigma$ certified accuracy and $A\tilde{CR}$, and therefore is, according to Definition 1, generally superior to the Isotropic DD one.

## 7.2 GENERALIZED CROSS-POLYTOPE CERTIFICATION ($\ell_1$ AND $\ell_1^\Lambda$-NORM CERTIFICATES)

To investigate $\ell_1$-ball vs. $\ell_1^\Lambda$-generalized cross-polytope certification via Uniform smoothing, we compare ANCER to the $\ell_1$ state-of-the-art results from RS4A (Yang et al., 2020). While the authors of the original work report best certified accuracy based on 15 networks trained at different $\sigma$ levels between 0.15 and 3.5 on CIFAR-10 (WideResNet40) and ImageNet (ResNet50) and due to limited computational resources, we perform the analysis on a subset of those networks with $\sigma = \{0.25, 0.5, 1.0\}$. We reproduce the results in Yang et al. (2020) as closely as possible, with details of the training procedure presented in Appendix E.2.

Table 2: Comparison of top-1 certified accuracy at different $\ell_1$ radii, $\ell_1$ average certified radius ($ACR$) and $\ell_1^\Lambda$ average certified proxy radius ($AC\tilde{R}$) obtained by using the isotropic $\sigma$ used for training the networks (Fixed $\sigma$); the isotropic data-dependent (Isotropic DD) optimization scheme from Alfarra et al. (2020); and ANCER's data-dependent anisotropic optimization.

| **CIFAR-10** | Certification | Accuracy @ $\ell_1$ radius (%) | | | | | | | $\ell_1\ ACR$ | $\ell_1^\Lambda\ AC\tilde{R}$ |
| | | 0.0 | 0.25 | 0.5 | 0.75 | 1.0 | 1.5 | 2.0 | | |
|---|---|---|---|---|---|---|---|---|---|---|
| RS4A Yang et al. (2020) | Fixed $\sigma$ | 92 | 83 | 75 | 71 | 46 | 0 | 0 | 0.775 | 0.775 |
| | Isotropic DD | 92 | 89 | 82 | 76 | 58 | 6 | 2 | 0.946 | 0.946 |
| | ANCER | 92 | 90 | 84 | 80 | 63 | 6 | 2 | **0.980** | **1.104** |
| **ImageNet** | | | | | | | | | | |
| RS4A Yang et al. (2020) | Fixed $\sigma$ | 78 | 73 | 67 | 63 | 0 | 0 | 0 | 0.683 | 0.683 |
| | Isotropic DD | 79 | 76 | 70 | 65 | 46 | 0 | 0 | 0.729 | 0.729 |
| | ANCER | 78 | 76 | 70 | 66 | 48 | 0 | 0 | **0.730** | **1.513** |

Figure 3 shows the top-1 certified accuracy as a function of the $\ell_1$ radius (top) and of the $\ell_1^\Lambda$-norm proxy radius (bottom) for RS4A, and Table 2 shows an overview of the certified accuracy at various $\ell_1$ radii, as well as $\ell_1\ ACR$ and $\ell_1^\Lambda\ AC\tilde{R}$.

As with the ellipsoid case, we notice that ANCER outperforms both Fixed $\sigma$ and Istropic DD for most $\ell_1$ radii, establishing new state-of-the-art results in CIFAR-10 at radii 0.5 and 1.0, and ImageNet at radii 0.5 (compared to previous results reported in Yang et al. (2020)). Once more and as expected, ANCER significantly improves the $\ell_1^\Lambda\ AC\tilde{R}$ for all radii, pointing to substantially larger certificates than the isotropic case. These results also establish that ANCER certifies the $\ell_1$-ball region obtained by previous work, in addition to the larger region obtained by the $\ell_1^\Lambda$ certificate, and thus we can consider it superior (with respect to Definition 1) to Isotropic DD.

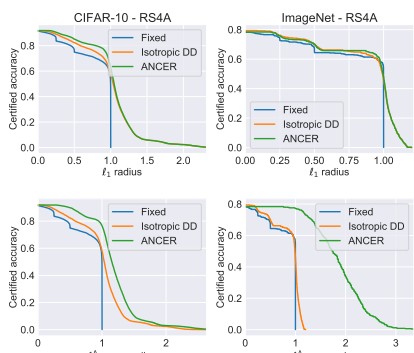

Figure 3: Distribution of top-1 certified accuracy as a function of $\ell_1$ radius (top) and $\ell_1^\Lambda$-norm proxy radius (bottom) obtained by different certification methods on CIFAR-10 and ImageNet.

### 7.3 CERTIFICATION RUNTIME

The certification procedures of Isotropic DD and ANCER tradeoff improved certified accuracy for runtime, since they require a sample-wise optimization to be run prior to the CERTIFY step described in Cohen et al. (2019), and a memory-based step as per Alfarra et al. (2020). The runtime of the optimization and certification procedures is roughly equal for $\ell_1$, $\ell_2$, $\ell_2^\Sigma$ and $\ell_1^\Lambda$ certification, and mostly depends on network architecture. As such, we report the average certification runtime for a test set sample

Table 3: Average certification time for each sample per architecture used: (a) ResNet18 ($\ell_2$, $\ell_2^\Sigma$ on CIFAR-10), (b) WideResNet40 ($\ell_1$, $\ell_1^\Lambda$ on CIFAR-10), and (c) ResNet50 (ImageNet).

| | Fixed $\sigma$ | Isotropic DD | ANCER |
|---|---|---|---|
| (a) | 1.6s | 1.8s | 2.7s |
| (b) | 7.4s | 9.5s | 11.5s |
| (c) | 109.5s | 136.0s | 147.0s |

on an NVIDIA Quadro RTX 6000 GPU for Fixed $\sigma$, Isotropic DD and ANCER (including the isotropic initialization step) in Table 3. We observe that the overall run time overhead for ANCER is not significant as compared to its certification gains.

## 8 CONCLUSION

In this work, we lay the theoretical foundations for anisotropic certification through a simple analysis, propose a metric for comparing general robustness certificates, and introduce ANCER, a certification procedure that estimates the parameters of the anisotropic smoothing distribution to maximize the certificate. Our experiments show that ANCER achieves state-of-the-art $\ell_1$ and $\ell_2$ certified accuracy on CIFAR-10 and ImageNet. Our anisotropic analysis enables further insights about the boundary of the safe region around an input sample, as compared to its isotropic counterpart.

## 9  REPRODUCIBILITY STATEMENT

We provide complete proofs for each of the theoretical results presented in Section 4 in Appendix B. Details on the practical implementation of the ANCER optimization algorithm is presented in Appendix C, while the memory-based procedure is detailed in Appendix D. An overview of the experimental setup used to obtain the results in Section 7 can be found at the top of that section, while details on the hyperparameters and network training are presented in Appendix E. We include source code in Python and instructions to reproduce our results as part of the supplementary material.

## 10  ETHICS STATEMENT

We confirm that no results in this paper involved studies on human subjects, and all experiments used open-source datasets.

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

# A QUALITATIVE MOTIVATION OF ANISOTROPIC CERTIFICATION

## A.1 VISUALIZING CIFAR-10 OPTIMIZED ISOTROPIC VS. ANISOTROPIC CERTIFICATES

To extend the illustration in Figure 1 to a higher dimensional input, we now analyze an example of the isotropic $\ell_2$ certification of randomized smoothing with $\mathcal{N}(0, \sigma^2 I)$, where $\sigma$ is optimized per input Alfarra et al. (2020), against ANCER, certifying an anisotropic region characterized by a diagonal $\ell_2^\Sigma$-norm. To do so, we consider a CIFAR-10 Krizhevsky (2009) dataset point $x$, where the input is of size (32x32x3). We perform the 2D analysis by considering the regions closest to a decision boundary. To do so, and following Moosavi-Dezfooli et al. (2019), we compute the Hessian of $f^y(x)$ with respect to $x$ where $y$ is the true label for $x$ with $f$ classifying $x$ correctly, *i.e.* $y = \arg\max_i f^i(x)$. In addition to the Hessian, we also compute its eigenvector decomposition, yielding the eigenvectors $\{\nu_i\}, i \in \{1, \ldots, 3072\}$ ordered in descending order of the absolute value of the respective eigenvalues. In Figure 4a, we show the

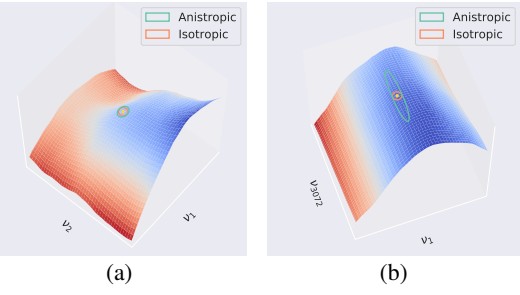

(a)       (b)

Figure 4: Illustration of the landscape of $f^y$ for points around an input point $x$, and two projections of an isotropic $\ell_2$ certified region and an anisotropic $\ell_2^\Sigma$2-norm region on a CIFAR-10 dataset example to a subset of two eigenvectors of the Hessian of $f^y$ (blue regions correspond to a higher confidence in $y$).

projection of the landscape of $f^y$ in the highest curvature directions, *i.e.* $\nu_1$ and $\nu_2$. Note that the isotropic certification, much as in Figure 1c, in these 2 dimensions is nearly optimal when compared to the anisotropic region. However, if we take the same projection with respect to the eigenvectors with the lowest and highest eigenvalues, *i.e.* $\nu_1$ and $\nu_{3072}$, the advantages of the anisotropic certification become clear as shown in Figure 4b.

## A.2 VISUALIZING SAFE IMAGES IN OPTIMIZED ANISOTROPIC CERTIFICATES

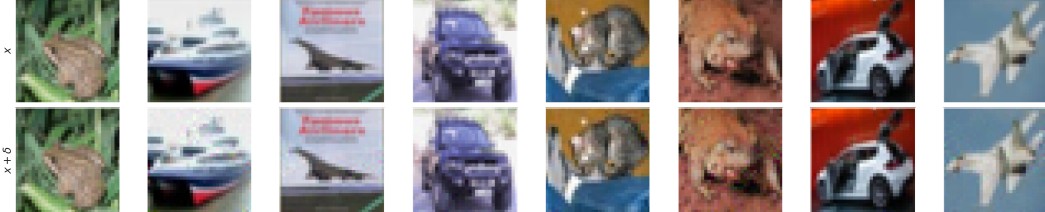

Figure 5: Visualization of a CIFAR-10 image $x$ and an example $x + \delta$ of an imperceptible change that *is not* inside the optimal isotropic certified region, but *is* covered by the anisotropic certificate.

As observed from the examples in Section 3 and Appendix A.1, anisotropic certification reasons more closely about the shape of the decision boundaries, allowing for further insights into constant prediction (safe) directions. In Figure 5, we present a series of test set images $x$, as well as practically indistinguishable $x + \delta$ images which *are not inside* the optimal certified isotropic $\ell_2$-balls for each input sample, yet *are within* the anisotropic certified regions. This showcases the merits of using anisotropic certification for characterizing larger safe regions.

# B ANISOTROPIC CERTIFICATION AND EVALUATION PROOFS

**Proposition 1** (restatement). *Consider a differentiable function $g : \mathbb{R}^n \to \mathbb{R}$. If $\sup_x \|\nabla g(x)\|_* \leq L$ where $\| \cdot \|_*$ has a dual norm $\|z\| = \max_x z^\top x$ s.t. $\|x\|_* \leq 1$, then $g$ is $L$-Lipschitz under norm $\| \cdot \|_*$, that is $|g(x) - g(y)| \leq L\|x - y\|$.*

*Proof.* Consider some $x, y \in \mathbb{R}^n$ and a parameterization in $t$ as $\gamma(t) = (1 - t)x + ty \ \forall t \in [0, 1]$. Note that $\gamma(0) = x$ and $\gamma(1) = y$. By the Fundamental Theorem of Calculus we have:

$$|g(y) - g(x)| = |g(\gamma(1)) - g(\gamma(0))| = \left| \int_0^1 \frac{dg(\gamma(t))}{dt} dt \right| = \left| \int_0^1 \nabla g^\top \nabla \gamma dt \right| \leq \int_0^1 \left| \nabla g^\top \nabla \gamma \right| dt$$

$$\leq \int_0^1 \|\nabla g(x)\|_* \|\nabla \gamma(t)\| dt \leq L \|y - x\|$$

$\square$

**Theorem 1** (restatement). *Let $g : \mathbb{R}^n \to \mathbb{R}^K$, $g^i$ be L-Lipschitz continuous under norm $\|\cdot\|_*$ $\forall i \in \{1, \ldots, K\}$, and $c_A = \arg\max_i g^i(x)$. Then, we have $\arg\max_i g^i(x + \delta) = c_A$ for all $\delta$ satisfying:*

$$\|\delta\| \leq \frac{1}{2L} \left( g^{c_A}(x) - \max_c g^{c \neq c_A}(x) \right).$$

*Proof.* Take $c_B = \arg\max_c g^{c \neq c_A}(x)$. By Proposition 1, we get:

$$|g^{c_A}(x + \delta) - g^{c_A}(x)| \leq L\|\delta\| \implies g^{c_A}(x + \delta) \geq g^{c_A}(x) - L\|\delta\|$$
$$|g^{c_B}(x + \delta) - g^{c_B}(x)| \leq L\|\delta\| \implies g^{c_B}(x + \delta) \leq g^{c_B}(x) + L\|\delta\|$$

By subtracting the inequalities and re-arranging terms, we have that as long as $g^{c_A}(x) - L\|\delta\| > g^{c_B}(x) + L\|\delta\|$, *i.e.* the bound in the Theorem, then $g^{c_A}(x + \delta) > g^{c_B}(x + \delta)$, completing the proof. $\square$

**Proposition 2** (restatement). *Consider $g_\Sigma(x) = \mathbb{E}_{\epsilon \sim \mathcal{N}(0, \Sigma)}[f(x + \epsilon)]$. $\Phi^{-1}(g_\Sigma(x))$ is 1-Lipschitz (i.e. $L = 1$) under the $\|\cdot\|_{\Sigma^{-1}, 2}$ norm.*

*Proof.* To prove Proposition 2, one needs to show that $\Phi^{-1}(g_\Sigma^i(x)) \ \forall i$ is 1-Lipschitz under the $\|\cdot\|_{\Sigma^{-1}, 2}$ norm. For ease of notation, we drop the superscript $g_\Sigma^i$ and use only $g$. We want to show that $\|\nabla \Phi^{-1}(g_\Sigma(x))\|_{\Sigma^{-1}, 2} = \|\Sigma^{1/2} \nabla \Phi^{-1}(g_\Sigma(x))\|_2 \leq 1$. Following the argument presented in Salman et al. (2019a), it suffices to show that, for any unit norm direction $u$ and $p = g_\Sigma(x)$, we have:

$$u^\top \Sigma^{\frac{1}{2}} \nabla g_\Sigma(x) \leq \frac{1}{\sqrt{2\pi}} \exp\left( -\frac{1}{2} (\Phi^{-1}(p))^2 \right). \tag{2}$$

We start by noticing that:

$$u^\top \Sigma^{\frac{1}{2}} \nabla g_\Sigma(x) = \frac{1}{(\sqrt{2\pi})^n \sqrt{|\Sigma|}} \int_{\mathbb{R}^n} f(t) u^\top \Sigma^{\frac{1}{2}} \Sigma^{-1} (t - x) \exp\left( -\frac{1}{2} (x - t) \Sigma^{-1} (x - t) \right) d^n t$$

$$= \mathbb{E}_{s \sim \mathcal{N}(0, \mathbf{I})}[f(x + \Sigma^{\frac{1}{2}} s) u^\top s] = \mathbb{E}_{v \sim \mathcal{N}(0, \Sigma)}[f(x + v) u^\top \Sigma^{-\frac{1}{2}} v].$$

We now need to find the optimal $f^* : \mathbb{R}^n \to [0, 1]$ that satisfies $g_\Sigma(x) = \mathbb{E}_{v \sim \mathcal{N}(0, \Sigma)}[f(x + v)] = p$ while maximizing the left hand size $\mathbb{E}_{v \sim \mathcal{N}(0, \Sigma)}[f(x + v) u^\top \Sigma^{-\frac{1}{2}} v]$. We argue that the maximizer is the following function:

$$f^*(x + v) = \mathbb{1} \left\{ u^\top \Sigma^{-\frac{1}{2}} v \geq -\Phi^{-1}(p) \right\}.$$

To prove that $f^*$ is indeed the optimal maximizer, we first show feasibility. **(i)**: It is clear that $f^* : \mathbb{R}^n \to [0, 1]$. **(ii)** Note that:

$$\mathbb{E}_{v \sim \mathcal{N}(0, \Sigma))} \left[ \mathbb{1} \left\{ u^\top \Sigma^{-\frac{1}{2}} v \geq -\Phi^{-1}(p) \right\} \right] = \mathbb{P}_{x \sim \mathcal{N}(0,1)} (x \geq -\Phi^{-1}(p)) = 1 - \Phi(-\Phi^{-1}(p)) = p.$$

To show the optimality of $f^*$, we show that it attains the right upper bound:

$$\mathbb{E}_{v \sim \mathcal{N}(0, \Sigma))} \left[ u^\top \Sigma^{-\frac{1}{2}} v \mathbb{1} \left\{ u^\top \Sigma^{-\frac{1}{2}} v \geq -\Phi^{-1}(p) \right\} \right] = \mathbb{E}_{x \sim \mathcal{N}(0,1)} \left[ x \mathbb{1} \left\{ x \geq -\Phi^{-1}(p) \right\} \right]$$

$$= \frac{1}{\sqrt{2\pi}} \int_{-\Phi^{-1}(p)}^\infty x \exp\left( -\frac{1}{2} x^2 \right) dx$$

$$= \frac{1}{\sqrt{2\pi}} \exp\left( -\frac{1}{2} (\Phi^{-1}(p))^2 \right)$$

obtaining the bound from Equation (2), and thus completing the proof. $\square$

**Proposition 3** (restatement). *Consider $g_\Lambda(x) = \mathbb{E}_{\epsilon \sim \mathcal{U}[-1,1]^n}[f(x + \Lambda\epsilon)]$. The classifier $g_\Lambda^i$ $\forall i$ is $1/2$-Lipschitz (i.e. $L = 1/2$) under the $\|\Lambda x\|_\infty$ norm.*

*Proof.* We begin by observing that the dual norm of $\|x\|_{\Lambda,1} = \|\Lambda^{-1}x\|_1$ is $\|x\|_* = \|\Lambda x\|_\infty$, since:

$$\max_{\|\Lambda^{-1}x\|_1 \le 1} x^\top y = \max_{\|z\|_1 \le 1} y^\top \Lambda z = \|\Lambda y\|_\infty.$$

Without loss of generality, we analyze $\partial g^i/\partial x_1$. Let $\hat{x} = [x_2, \ldots, x_n] \in \mathbb{R}^{n-1}$, then:

$$\frac{\lambda_1 \partial g^i}{\partial x_1} = \frac{\lambda_1}{(2\lambda)^n} \frac{\partial}{\partial x_1} \int_{[-1,1]^{n-1}} \int_{-1}^1 f^i(x_1 + \lambda_1\epsilon_1, \hat{x} + \hat{\Lambda}\hat{\epsilon}) d\epsilon_1 d^{n-1}\hat{\epsilon}$$

$$= \frac{1}{2^n} \int_{[-1,1]^{n-1}} (f^i(x_1 + 1, \hat{x} + \hat{\Lambda}\hat{\epsilon}) - f^i(x_1 - 1, \hat{x} + \hat{\Lambda}\hat{\epsilon})) d^{n-1}\hat{\epsilon}$$

Thus,

$$\left| \frac{\lambda_1 \partial g^i}{\partial x_1} \right| \le \frac{1}{2^n \prod_{j=2}^n \lambda_j} \int_{[-1,1]^{n-1}} \left| f^i(x_1 + 1, \hat{x} + \hat{\Lambda}\hat{\epsilon}) - f^i(x_1 - 1, \hat{x} + \hat{\Lambda}\hat{\epsilon}) \right| d^{n-1}\hat{\epsilon} \le \frac{1}{2}.$$

The second and last steps follow by the change of variable $t = x_1 + \lambda_1\epsilon_1$ and Leibniz rule. Following a symmetric argument, $\left| \lambda_j \partial g^i/\partial x_j \right| \le 1/2$ $\forall i$ resulting in having $\|\Lambda \nabla g^i(x)\|_\infty = \max_i \lambda_i \left| \partial g^i/\partial x_i \right| \le 1/2$ $\forall i$ concluding the proof. $\qquad \square$

**Proposition 4** (restatement). $\mathcal{V}\left( \{\delta : \|\Lambda^{-1}\delta\|_1 \le r\} \right) = \frac{(2r)^n}{n!} \prod_i \lambda_i.$

*Proof.* Take $A = r\Lambda^{-1} = \text{diag}(1/r\lambda_1, \ldots, 1/r\lambda_n) = \text{diag}(a_1, \ldots, a_n)$.

We can re-write the region as $\{x : \sum_i a_i |x_i| \le 1\}$, from which it is clear to see that this region is an origin centered, axis-aligned simplex with the set of vertices $\mathcal{V} = \{\pm 1/a_i \mathbf{e}_i\}_{i=1}^n$, where $\mathbf{e}_i$ is the standard basis vector $i$.

Define the sets of vertices $\mathcal{V}^t = \mathcal{V} \setminus \{-1/a_n \mathbf{e}_n\}$ and $\mathcal{V}^b = \mathcal{V} \setminus \{1/a_n \mathbf{e}_n\}$. Given the symmetry around the origin, each of these sets defines an $n$-dimensional *hyperpyramid* with a shared *base* $B_{n-1}$ given by the $n-1$-dimensional hyperplane defined by all vertices where $x_n = 0$, and an *apex* at the vertex $1/a_n \mathbf{e}_n$ (or $-1/a_n \mathbf{e}_n$ in the case of $\mathcal{V}^b$). The volume of each of these $n-1$-dimensional hyperpyramids is given by $\mathcal{V}(B_{n-1})/na_n$ (Kendall (2004)), yielding a total volume of $V_n = \frac{2}{n} \frac{1}{a_n} \mathcal{V}(B_{n-1})$. The same argument can be applied to compute $\mathcal{V}(B_{n-1})$ which is a union of two $n-1$-dimensional hyperpyramids. This forms a recursion that completes the proof. $\qquad \square$

*Proof.* (**Alternative Proof.**) We consider the case that $\Lambda^{-1}$ is a general positive definite matrix that is not necessarily diagonal. Note that $\mathcal{V}\left( \{\delta : \|\Lambda^{-1}\delta\|_1 \le r\} \right) = \mathcal{V}\left( \{\delta : \|(r\Lambda)^{-1}\delta\|_1 \le 1\} \right) = r^n |\Lambda| \mathcal{V}\left( \{\delta : \|\delta\|_1 \le 1\} \right)$ where $|r\Lambda|$ denotes the determinant. The last equality follows by the volume of a set under a linear map and noting that $\{\delta : \|(r\Lambda)^{-1}\delta\|_1 \le 1\} = \{r\Lambda\delta : \|\delta\|_1 \le r\}$. At last, $\{\delta : \|\delta\|_1 \le 1\}$ can be expressed as the disjoint union of $2^n$ simplexes. Thus, we have $\mathcal{V}\left( \{\delta : \|\Lambda^{-1}\delta\|_1 \le r\} \right) = (2r)^n/n! |\Lambda|$ since the volume of a simplex is $1/n!$ completing the proof. $\qquad \square$

For completeness, we supplement the previous result with bounds on the volume that may be useful for future readers.

**Proposition 5.** *For any positive definite $\Lambda^{-1} \in \mathbb{R}^{n \times n}$, we have the following:*

$$\left( \frac{2r}{n} \right)^n \mathcal{V}\left( \mathcal{Z}(\Lambda) \right) \le \mathcal{V}\left( \{\delta : \|\Lambda^{-1}\delta\|_1 \le r\} \right) \le (2r)^n \mathcal{V}\left( \mathcal{Z}(\Lambda) \right)$$

*where $\mathcal{V}\left( \mathcal{Z}(\Lambda) \right) = \sqrt{|\Lambda^\top \Lambda|}$ which is the volume of the zonotope with a generator matrix $\Lambda$.*

*Proof.* Let $S_1 = \{\delta : \|\Lambda^{-1}\delta\|_1 \le r\}$, $S_\infty = \{\delta : \|\Lambda^{-1}\delta\|_\infty \le r\}$ and $S_\infty^n = \{\delta : n\|\Lambda^{-1}\delta\|_\infty \le r\}$. Since $\|\Lambda^{-1}\delta\|_\infty \le \|\Lambda^{-1}\delta\|_1 \le n\|\Lambda^{-1}\delta\|_\infty$, then $S_\infty \supseteq S_1 \supseteq S_\infty^n$. Therefore, we have $\mathcal{V}(S_\infty) \ge \mathcal{V}(S_1) \ge \mathcal{V}(S_\infty^n)$. At last note that, $S_\infty^n = \{\frac{r}{n}\Lambda\delta : \|\delta\|_\infty \le 1\}$ and that with the change of variables $\delta = 2u - 1_n$ where $1_n$ is a vector of all ones, we have $S_\infty^n = \mathcal{Z}\left(\frac{2r}{n}\Lambda\right) \oplus \frac{-r}{n}\Lambda 1_n$ where $\oplus$ is a Minkowski sum and noting that $\frac{r}{n}\Lambda 1_n$ is a single point in $\mathbb{R}^n$. Therefore, $\mathcal{V}\left(\mathcal{Z}\left(\frac{2r}{n}\Lambda\right) \oplus \frac{-r}{n}\Lambda 1_n\right) = \left(2r/n\right)^n \mathcal{V}\left(\mathcal{Z}(\Lambda)\right)$. The upper bound follows with a similar argument completing the proof. $\qquad\square$

### B.1 CERTIFICATION UNDER GAUSSIAN MIXTURE SMOOTHING DISTRIBUTION

We consider a general, $K$-component, zero-mean Gaussian mixture smoothing distribution $\mathcal{G}$ such that:

$$\mathcal{G}(\{\alpha_i, \Sigma_i\}_{i=1}^K) := \sum_{i=1}^K \alpha_i \mathcal{N}(0, \Sigma_i), \quad \text{s.t.} \quad \sum_i \alpha_i = 1, 0 < \alpha_i \le 1 \tag{3}$$

Given $f$ and as per the recipe described in Section 4, we are interested in the Lipschitz constant of the smooth classifier $g_\mathcal{G}(x) = (f * \mathcal{G})(x) = \sum_i^K \alpha_i g_{\Sigma_i} = \sum_i^K \alpha_i(f * \mathcal{N}(0, \Sigma_i)) = \sum_i \alpha_i g_{\Sigma_i}(x)$ where $g_{\Sigma_i}$ is defined as in the Gaussian case.

Note the weaker bound when compared to Proposition 2, for each of the Gaussian components presented in the following proposition.

**Proposition 6.** $g_\Sigma$ *is* $\sqrt{2/\pi}$-*Lipschitz under* $\|.\|_{\Sigma^{-1},2}$ *norm.*

*Proof.* Following a similar argument to the proof of Proposition 2, we get:

$$u^\top \Sigma^{\frac{1}{2}} \nabla g_\Sigma(x) \le \frac{1}{(2\pi)^{n/2}\sqrt{|\Sigma|}} \int_{\mathbb{R}^n} |u^\top \Sigma^{-\frac{1}{2}}(t-x)| \exp\left(-\frac{1}{2}(x-t)^\top \Sigma^{-1}(x-t)\right) d^n t$$
$$= \mathbb{E}_{s\sim\mathcal{N}(0,\mathbf{I})}\left[|u^\top s|\right] = \mathbb{E}_{v\sim\mathcal{N}(0,1)}\left[|v|\right] = \sqrt{2/\pi}.$$

$\qquad\square$

With Proposition 6, we obtain a Lipschitz constant for a Gaussian mixture smoothing distribution as:

**Proposition 7.** $g_\mathcal{G}$ *is* $\sqrt{\pi/2}$-*Lipschitz under* $\|\delta\|_{\mathcal{B}^{-1},2}$ *norm, where* $\mathcal{B}^{-1} = \sum_i^K \alpha_i \Sigma_i^{-1}$.

*Proof.*

$$|g_\mathcal{G}(x+\delta) - g_\mathcal{G}(x)| \le \sum_i \alpha_i |g_{\Sigma_i}(x+\delta) - g_{\Sigma_i}(x)|$$

$$\le \sqrt{\frac{\pi}{2}} \sum_i \alpha_i \|\delta\|_{\Sigma_i,2} \le \sqrt{\frac{\pi}{2}} \sqrt{\delta^\top \left(\sum_i \alpha_i \Sigma_i^{-1}\right) \delta} = \sqrt{\frac{\pi}{2}} \|\delta\|_{\mathcal{B},2},$$

Obtained by first applying the triangle inequality, then Proposition 2 followed by Jensen's inequality.
$\qquad\square$

Thus yielding the following certificate by combining Proposition 7 and Theorem 1.

**Corollary 3.** *Let* $c_A = \arg\max_i g_\mathcal{G}(x)$, *then* $\arg\max_i g_\mathcal{G}^i(x+\delta) = c_A$ *for all* $\delta$ *satisfying:*

$$\|\delta\|_{\mathcal{B},2} \le \frac{1}{\sqrt{2\pi}}\left(g_\mathcal{G}^{c_A}(x) - \max_c g_\mathcal{G}^{c\ne c_A}(x)\right).$$

*where* $\mathcal{B}^{-1} = \sum_i^K \alpha_i \Sigma_i^{-1}$.

## C ANCER OPTIMIZATION

In this section we detail the implementation choices required to solving Equation (1). For ease of presentation, we restate the ANCER optimization problem (with $\Theta^x = \text{diag}(\{\theta_i^x\}_{i=1}^n)$):

$$\arg\max_{\Theta^x} \ r^p(x, \Theta^x) \sqrt[n]{\prod_i \theta_i^x} \qquad \text{s.t.} \quad \min_i \theta_i^x r^p(x, \Theta^x) \geq r_{\text{iso}}^*,$$

where $r^p(x, \Theta^x)$ is the gap value under the anisotropic smoothing distribution, and $r_{\text{iso}}^*$ is the optimal isotropic radius, i.e. $\bar{\theta}^x r^p(x, \bar{\theta}^x)$ for $\bar{\theta}^x \in \mathbb{R}^+$. This is a nonlinear constrained optimization problem that is challenging to solve. As such, we relax it, and solve instead:

$$\arg\max_{\Theta^x} \ r^p(x, \Theta^x) \sqrt[n]{\prod_i \theta_i^x} + \kappa \min_i \theta_i^x r^p(x, \Theta^x) \quad \text{s.t.} \quad \theta_i^x \geq \bar{\theta}^x$$

given a hyperparameter $\kappa \in \mathbb{R}^+$. While the constraint $\theta_i^x \geq \bar{\theta}^x$ is not explicitly required to enforce the *superset* condition over the isotropic case, it proved itself beneficial from an empirical perspective. To sample from the distribution parameterized by $\Theta^x$ (in our case, either a Gaussian or Uniform), we make use of the *reparameterization trick*, as in Alfarra et al. (2020). The solution of this optimization problem can be found iteratively by performing projected gradient ascent.

A standalone implementation for the ANCER optimization stage is presented in Listing 1, whereas the full code integrated in our code base is available as supplementary material. To perform certification, we simply feed the output of this optimization to the certification procedure from Cohen et al. (2019).

```python
import torch
from torch.autograd import Variable
from torch.distributions.normal import Normal

class Certificate():
    def compute_proxy_gap(self, logits: torch.Tensor):
        raise NotImplementedError

    def sample_noise(self, batch: torch.Tensor, repeated_theta: torch.Tensor):
        raise NotImplementedError

    def compute_gap(self, pABar: float):
        raise NotImplementedError

class L2Certificate(Certificate):
    def __init__(self, batch_size: int, device: str = "cuda:0"):
        self.m = Normal(torch.zeros(batch_size).to(device),
                        torch.ones(batch_size).to(device))
        self.device = device
        self.norm = "l2"

    def compute_proxy_gap(self, logits: torch.Tensor):
        return self.m.icdf(logits[:, 0].clamp_(0.001, 0.999)) - \
            self.m.icdf(logits[:, 1].clamp_(0.001, 0.999))

    def sample_noise(self, batch: torch.Tensor, repeated_theta: torch.Tensor):
        return torch.randn_like(batch, device=self.device) * repeated_theta

    def compute_gap(self, pABar: float):
        return norm.ppf(pABar)

class L1Certificate(Certificate):
    def __init__(self, device="cuda:0"):
        self.device = device
        self.norm = "l1"

    def compute_proxy_gap(self, logits: torch.Tensor):
        return logits[:, 0] - logits[:, 1]

    def sample_noise(self, batch: torch.Tensor, repeated_theta: torch.Tensor):
        return 2 * (torch.rand_like(batch, device=self.device) - 0.5) * repeated_theta

    def compute_gap(self, pABar: float):
        return 2 * (pABar - 0.5)
```

```python
def ancer_optimization(
        model: torch.nn.Module, batch: torch.Tensor,
        certificate: Certificate, learning_rate: float,
        isotropic_theta: torch.Tensor, iterations: int,
        samples: int, kappa: float, device: str = "cuda:0"):
    """Optimize batch using ANCER, assuming isotropic initialization point.

    Args:
        model: trained network
        batch: inputs to certify around
        certificate: instance of desired certification object
        learning_rate: optimization learning rate for ANCER
        isotropic_theta: initialization isotropic value per input in batch
        iterations: number of iterations to run the optimization
        samples: number of samples per input and iteration
        kappa: relaxation hyperparameter
    """
    batch_size = batch.shape[0]
    img_size = np.prod(batch.shape[1:])

    # define a variable, the optimizer, and the initial sigma values
    theta = Variable(isotropic_theta, requires_grad=True).to(device)
    optimizer = torch.optim.Adam([theta], lr=learning_rate)
    initial_theta = theta.detach().clone()

    # reshape vectors to have ``samples`` per input in batch
    new_shape = [batch_size * samples]
    new_shape.extend(batch[0].shape)
    new_batch = batch.repeat((1, samples, 1, 1)).view(new_shape)

    # solve iteratively by projected gradient ascend
    for _ in range(iterations):
        theta_repeated = theta.repeat(1, samples, 1, 1).view(new_shape)

        # Reparameterization trick
        noise = certificate.sample_noise(new_batch, theta_repeated)
        out = model(
            new_batch + noise
        ).reshape(batch_size, samples, -1).mean(dim=1)

        vals, _ = torch.topk(out, 2)
        gap = certificate.compute_proxy_gap(vals)

        prod = torch.prod(
            (theta.reshape(batch_size, -1))**(1/img_size), dim=1)
        proxy_radius = prod * gap

        radius_maximizer = - (
            proxy_radius.sum() +
            kappa *
            (torch.min(theta.view(batch_size, -1), dim=1).values*gap).sum()
        )
        radius_maximizer.backward()
        optimizer.step()

        # project to the initial theta
        with torch.no_grad():
            torch.max(theta, initial_theta, out=theta)

    return theta
```

Listing 1: Python implementation of the ANCER optimization routine using PyTorch Paszke et al. (2019)

## D  MEMORY-BASED CERTIFICATION FOR ANCER

To guarantee the soundness of the ANCER classifier, we use an adapted version of the data-dependent memory-based solution presented in Alfarra et al. (2020). The modified algorithm involves a post-processing certification step that obtains adjusted certification statistics based on the memory

procedure from Alfarra et al. (2020) (see the original paper for more details). We present an adapted version to ANCER of this post-processing memory-based step in Algorithm 1.

---

**Algorithm 1:** Memory-Based Certification

**Input:** input point $x_{N+1}$, certified region $\mathcal{R}_{N+1}$, prediction $\mathcal{C}_{N+1}$, and memory $\mathcal{M}$
**Result:** Prediction for $x_{N+1}$ and certified region at $x_{N+1}$ that does not intersect with any
       certified region in $\mathcal{M}$.
**for** $(x_i, \mathcal{C}_i, \mathcal{R}_i) \in \mathcal{M}$ **do**
    **if** $\mathcal{C}_{N+1} \neq \mathcal{C}_i$ **then**
        **if** $x_{N+1} \in \mathcal{R}_i$ **then**
            **return** ABSTAIN, 0
        **else if** *MaxIntersect($\mathcal{R}_{N+1}, \mathcal{R}_i$) and Intersect($\mathcal{R}_{N+1}, \mathcal{R}_i$)* **then**
            $\mathcal{R}'_{N+1}$ = LargestOutSubset($\mathcal{R}_i, \mathcal{R}_{N+1}$);
            $\mathcal{R}_{N+1} \leftarrow \mathcal{R}'_{N+1}$;
**end**
add $(x_{N+1}, \mathcal{C}_{N+1}, \mathcal{R}_{N+1})$ to $\mathcal{M}$;
**return** $\mathcal{C}_{N+1}, \mathcal{R}_{N+1}$;

---

Note that the proposed certified region $\mathcal{R}_{N+1}$ emerges from our certification bounds presented in Sections 4.1 and 4.2. There are a few differences between our proposed Algorithm 1 with respect to the original variant presented in Alfarra et al. (2020). The first is that we remove the computation of the largest certifiable subset of a certified region $\mathcal{R}_{N+1}$ when there exists an $i$ such that $x_{N+1} \in \mathcal{R}_i$ with a different class prediction, *i.e.* (LargestInSubset in Alfarra et al. (2020)) due to the complexity of the operation in the anisotropic case. As an example, it is generally difficult to find the largest volume ellipsoid contained in another ellipsoid. Due to this complexity, we choose to simply ABSTAIN instead. Given the high dimensionality of the data, empirically, we never found a certificate in this situation within our experiments. Further, to ease the computational burden of the Intersect function, we introduce and instantiate the function MaxIntersect first which checks whether the $\ell_p$-ball over-approximation of the region $\mathcal{R}_{N+1}$ intersects with a $\ell_p$ over-approximation of $\mathcal{R}_i$. This follows since when the $\ell_p$ balls over-approximation to the anisotropic regions $\mathcal{R}_{N+1}$ and $\mathcal{R}_i$ do not intersect, then $\mathcal{R}_{N+1}$ and $\mathcal{R}_i$ do not intersect either. Only in cases in which those over-approximation regions intersect, we run the more expensive Intersect procedure. We present practical implementations for MaxIntersect, Intersect and LargestOutSubset for the ellipsoids and generalized cross-polytopes considered in this paper.

### D.1 IMPLEMENTING MAXINTERSECT($\mathcal{R}_{\mathbf{A}}$, $\mathcal{R}_{\mathbf{B}}$) IN THE ELLIPSOID AND GENERALIZED CROSS-POLYTOPE CASES

Given the two regions $\mathcal{R}_{\mathbf{A}}$ and $\mathcal{R}_{\mathbf{B}}$, consider $\ell_p$-ball approximations of those regions, $\mathcal{R}_{\tilde{\mathbf{B}}} = \{x \in \mathbb{R}^n : \|x - a\|_p \leq r_a\}$ and $\mathcal{R}_{\tilde{\mathbf{B}}} = \{x \in \mathbb{R}^n : \|x - b\|_p \leq r_b\}$ such that $\mathcal{R}_{\mathbf{A}} \subseteq \mathcal{R}_{\tilde{\mathbf{A}}}$ and $\mathcal{R}_{\mathbf{B}} \subseteq \mathcal{R}_{\tilde{\mathbf{B}}}$.

**Lemma 1.** *If $\|a - b\|_p > r_a + r_b$, then $\mathcal{R}_{\mathbf{A}} \cap \mathcal{R}_{\mathbf{B}} = \emptyset$.*

*Proof.* For the sake of contradiction, let $\|a - b\|_p > r_a + r_b$ and $x \in \mathcal{R}_{\tilde{\mathbf{A}}} \cap \mathcal{R}_{\tilde{\mathbf{B}}}$. Then, we have that $\|x - a\| \leq r_a$ and $\|x - b\| \leq r_b$. However:

$$r_a + r_b < \|a - b\|_p \leq \|x - a\|_p + \|x - b\|_p \leq r_a + r_b,$$

forming a contradiction. Thus, $\mathcal{R}_{\tilde{\mathbf{A}}} \cap \mathcal{R}_{\tilde{\mathbf{B}}} = \emptyset$, which in turn implies $\mathcal{R}_{\mathbf{A}} \cap \mathcal{R}_{\mathbf{B}} = \emptyset$ since $\mathcal{R}_{\mathbf{A}}$ and $\mathcal{R}_{\mathbf{B}}$ are subsets of $\mathcal{R}_{\tilde{\mathbf{A}}}$ and $\mathcal{R}_{\tilde{\mathbf{B}}}$, respectively. $\qquad\square$

This forms a fast, maximum intersection check for ellipsoids, *i.e.* $p = 2$, and generalized cross-polytopes, *i.e.* $p = 1$. The MaxIntersect function returns False if $\|a - b\|_p > r_a + r_b$, and True otherwise.

### D.2 IMPLEMENTING INTERSECT($\mathcal{R}_{\mathbf{A}}, \mathcal{R}_{\mathbf{B}}$) IN THE ELLIPSOID CASE

The problem of efficiently checking if two ellipsoids intersect is not trivial. We rely on the work of Ros et al. (2002); Gilitschenski & Hanebeck (2012) with missing proofs from Gilitschenski & Hanebeck (2012) for completeness.

**Lemma 2.** *Let $\mathcal{R}_{\mathbf{A}} = \{x \in \mathbb{R}^n : (x-a)^\top \mathbf{A}(x-a) \leq 1\}$ and $\mathcal{R}_{\mathbf{B}} = \{x \in \mathbb{R}^n : (x-b)^\top \mathbf{B}(x-b) \leq 1\}$ define two ellipsoids centered at $a$ and $b$, respectively. We have that $\mathcal{R} = \{x : t(x-a)^\top \mathbf{A}(x-a) + (1-t)(x-b)^\top \mathbf{B}(x-b) \leq 1\}$ for any $t \in [0,1]$ satisfies $\mathcal{R}_{\mathbf{A}} \cap \mathcal{R}_{\mathbf{B}} \subseteq \mathcal{R} \subseteq \mathcal{R}_{\mathbf{A}} \cup \mathcal{R}_{\mathbf{B}}$.*

*Proof.* By considering the convex combination of the left-hand side of the inequalities defining the regions $\mathcal{R}_{\mathbf{A}}$ and $\mathcal{R}_{\mathbf{B}}$, it becomes obvious that $x \in \mathcal{R}_{\mathbf{A}} \cap \mathcal{R}_{\mathbf{B}} \implies x \in \mathcal{R}$, concluding the left side of the property. As for the right side, it suffices to show that if $x \notin \mathcal{R}_{\mathbf{A}}$ and $x \in \mathcal{R}$ then $x \in \mathcal{R}_{\mathbf{B}}$ and, similarly, that if $x \notin \mathcal{R}_{\mathbf{B}}$ and $x \in \mathcal{R}$ then $x \in \mathcal{R}_{\mathbf{A}}$. We show the first case since the second follows by symmetry. Without loss of generality, we assume that $a = b = \mathbf{0}_n$. Now, let $x$ be such that $x^\top \mathbf{A} x > 1$ and $tx^\top \mathbf{A} x + (1-t)x^\top \mathbf{B} x \leq 1$ since $x \notin \mathcal{R}_{\mathbf{A}}$ and $x \in \mathcal{R}$. Then, since $x \in \mathcal{R}$, we have that $(1-t)x^\top \mathbf{B} x \leq 1 - tx^\top \mathbf{A} x \leq 1$ since $x^\top \mathbf{A} x > 1$ which implies that $x \in \mathcal{R}_{\mathbf{B}}$. $\square$

Note that the previous result holds without loss of generality when for the radius 1 as the radius can be absorbed in $\mathbf{A}$ and $\mathbf{B}$. As the following Lemma was shown by Gilitschenski & Hanebeck (2012) without proof, we complement it below for completeness.

**Lemma 3.** *The set $\mathcal{R}$ is equivalent to the following ellipsoid $\mathcal{R} = \{x : (x-m)^\top \mathbf{E}_t(x-m) \leq K(t)\}$ where $\mathbf{E}_t = t\mathbf{A} + (1-t)\mathbf{B}$, $m = \mathbf{E}_t^{-1}(t\mathbf{A}a + (1-t)\mathbf{B}b)$, and $K(t) = 1 - ta^\top \mathbf{A}a - (1-t)b^\top \mathbf{B}b + m^\top \mathbf{E}_t m$.*

*Proof.*

$$t(x-a)^\top \mathbf{A}(x-a) + (1-t)(x-b)^\top \mathbf{B}(x-b) \leq 1$$
$$\Leftrightarrow x^\top \underbrace{(t\mathbf{A} + (1-t)\mathbf{B})}_{\mathbf{E}_t} x - 2x^\top \underbrace{(t\mathbf{A}a + (1-t)\mathbf{B}b)}_{\mathbf{E}_t m} \leq 1 - ta^\top \mathbf{A}a - (1-t)b^\top \mathbf{B}b$$
$$\Leftrightarrow (x-m)^\top \mathbf{E}_t(x-m) \leq 1 - ta^\top \mathbf{A}a - (1-t)b^\top \mathbf{B}b + m^\top \mathbf{E}_t m$$

The last equality follows by adding and subtracting $m^\top \mathbf{E}_t m$ and concluding the proof. $\square$

**Proposition 8.** *The set of points satisfying $\mathcal{R}$ for $t \in (0,1)$ is either an empty set, a single point, or the ellipsoid $\mathcal{R}$.*

*Proof.* We first observe that since $\mathbf{A}$ and $\mathbf{B}$ are positive definite, then $\mathbf{E}_t$ is positive definite. Then observe that for a choice of $t \in (0,1)$ such that $K(t) < 0$, the set $\mathcal{R}$ is an empty set, and since $\mathcal{R} \supseteq \mathcal{R}_{\mathbf{A}} \cap \mathcal{R}_{\mathbf{B}}$, the two sets do not intersect. If $K(t) = 0$, then the only point satisfying $\mathcal{R}$ is the center at $m$. Following a similar argument, then the two ellipsoids intersect at a point. At last for a choice of $t$ such that $K(t) > 0$, then $\mathcal{R}$ defines an ellipsoid. $\square$

As per Theorem 8, it suffices to find some $t \in [0,1]$ under which $K(t) < 0$ to guarantee that the ellipsoids do not intersect. To that end, we solve the following convex optimization problem: $t^* = \operatorname{argmin}_{t \in [0,1]} K(t)$ and check the condition if $K(t^*) < 0$. Moreover, as shown by Ros et al. (2002); Gilitschenski & Hanebeck (2012) $K(t)$ is convex in the domain $t \in (0,1)$. With several algebraic manipulations, one can show that $K(t)$ has the following equivalent forms:

$$K(t) = 1 - ta^\top \mathbf{A}a - (1-t)b^\top \mathbf{B}b + m^\top \mathbf{E}_t m$$
$$K(t) = 1 - t(1-t)(b-a)^\top \mathbf{B}\mathbf{E}_t^{-1}\mathbf{A}(b-a)$$
$$K(t) = 1 - (b-a)^\top \left(\frac{1}{1-t}\mathbf{B}^{-1} + \frac{1}{t}\mathbf{A}^{-1}\right)^{-1}(b-a)$$

Observe that for ANCER, we have that both $\mathbf{A}$ and $\mathbf{B}$ to be diagonals with diagonal elements $\{\mathbf{A}_{ii}\}_{i=1}^n$ and $\{\mathbf{B}_{ii}\}_{i=1}^n$, respectively, resulting in the following simple form for $K(t)$:

$$K(t) = 1 - \sum_{i=1}^n (b_i - a_i)^2 \frac{t(1-t)\mathbf{A}_{ii}\mathbf{B}_{ii}}{t\mathbf{A}_{ii} + (1-t)\mathbf{B}_{ii}}.$$

The `Intersect` function in the ellipsoid case returns `False` if there exists a $t \in (0,1)$ such that $K(t) < 0$, *i.e.* ellipsoids do not intersect, and `True` otherwise.

### D.3 Implementing Intersect($\mathcal{R}_\mathbf{A}, \mathcal{R}_\mathbf{B}$) in the Generalized Cross-Polytope Case

Let $\mathcal{R}_\mathbf{A}$ and $\mathcal{R}_\mathbf{B}$ be two generalized cross-polytopes $\mathcal{R}_\mathbf{A} = \{x \in \mathbb{R}^n : \|\mathbf{A}(x - a)\|_1 \leq 1\}$ and $\mathcal{R}_\mathbf{B} = \{x \in \mathbb{R}^n : \|\mathbf{B}(x - b)\|_1 \leq 1\}$, where $\mathbf{A}$ and $\mathbf{B}$ are positive definite diagonal matrices with elements $\{\mathbf{A}_{ii}\}_{i=1}^n$ and $\{\mathbf{B}_{ii}\}_{i=1}^n$, respectively. We are interested in deciding whether $\mathcal{R}_\mathbf{A}$ and $\mathcal{R}_\mathbf{B}$ intersect. However, given the conservative context in which Intersect is used in Algorithm 1, we only need to make sure that the function only returns False if it is guaranteed that $\mathcal{R}_\mathbf{A} \cap \mathcal{R}_\mathbf{B} = \emptyset$.

As such, we are able to simplify the complex problem of generalized cross-polytope intersection to the much simpler one of ellipsoid over-approximation intersection. We do this by considering the over-approximation, *i.e.* superset, ellipsoids $\mathcal{R}_{\tilde{\mathbf{A}}} = \{x \in \mathbb{R}^n : \|\mathbf{A}(x - a)\|_2 \leq 1\}$ and $\mathcal{R}_{\tilde{\mathbf{B}}} = \{x \in \mathbb{R}^n : \|\mathbf{B}(x - b)\|_2 \leq 1\}$, and perform the ellipsoid intersection check presented in Appendix D.2. If $\mathcal{R}_{\tilde{\mathbf{A}}} \cap \mathcal{R}_{\tilde{\mathbf{B}}} = \emptyset$, then this implies that $\mathcal{R}_\mathbf{A} \cap \mathcal{R}_\mathbf{B} = \emptyset$ and we can safely return False. Otherwise, we conservatively assume the generalized cross-polytopes intersect, and return True, triggering the reduction procedure detailed in Appendix D.5.

### D.4 Implementing LargestOutSubset($\mathcal{R}_\mathbf{A}, \mathcal{R}_\mathbf{B}$) in the Ellipsoid Case

Given two ellipsoids $\mathcal{R}_\mathbf{A} = \{x \in \mathbb{R}^n : (x - a)^\top \mathbf{A}(x - a) \leq 1\}$ and $\mathcal{R}_\mathbf{B} = \{x \in \mathbb{R}^n : (x - b)^\top \mathbf{B}(x - b) \leq 1\}$ that do intersect where $\mathbf{A}$ and $\mathbf{B}$ are positive definite diagonal matrices, the task is to find the largest possible ellipsoid $\mathcal{R}_{\tilde{\mathbf{B}}}$ centered at $b$ such that $\mathcal{R}_{\tilde{\mathbf{B}}} \subseteq \mathcal{R}_\mathbf{B}$ where $\mathcal{R}_\mathbf{A} \cap \mathcal{R}_{\tilde{\mathbf{B}}} = \emptyset$.

Finding a maximum ellipsoid that satisfies those conditions is not trivial, so instead we consider a maximum enclosing $\ell_2$-ball of $\mathcal{R}_\mathbf{B}$, $\mathcal{R}_{\tilde{\mathbf{B}}} = \{x \in \mathbb{R}^n : \|x - b\|_2 \leq r\}$, that does not intersect $\mathcal{R}_\mathbf{A}$. To obtain this ball, we project the center of $\mathcal{R}_\mathbf{B}$, $b$, to the ellipsoid $\mathcal{R}_\mathbf{A}$. Particularly, we formulate the problem as the projection of a vector $y = b - a$ onto an ellipsoid with the same shape as $\mathcal{R}_\mathbf{A}$ centered at $\mathbf{0}_n$. This is equivalent to solving the following optimization problem for a symmetric positive definite matrix $\mathbf{A}$:

$$\min_x \frac{1}{2} \|x - y\|_2^2 \qquad \text{s.t.} \quad x^\top \mathbf{A} x \leq 1.$$

Note that the objective function is convex, and the constraint forms a convex set. Forming the Lagrangian to this problem, we obtain:

$$\mathcal{L}(x, \lambda) = \frac{1}{2} \|x - y\|_2^2 + \lambda \left(x^\top \mathbf{A} x - 1\right),$$

where $\lambda > 0$. Therefore, the global optimal solution must satisfy the KKT conditions below:

$$\frac{\partial \mathcal{L}}{\partial x} = 0 \rightarrow x^* = (2\lambda \mathbf{A} + I)^{-1} y,$$

$$\frac{\partial \mathcal{L}}{\partial \lambda} = 0 \rightarrow \underbrace{y^\top (2\lambda \mathbf{A} + I)^{-\top} \mathbf{A} (2\lambda \mathbf{A} + I)^{-1} y - 1}_{f(\lambda)} = 0.$$

Thus, to project the vector $y$ on our region the ellipsoid characterized by $\mathbf{A}$, one needs to solve the scalar optimization $f(\lambda) = 0$ then substitute back in the formula of $x^*$. Further, given $\mathbf{A} = \text{diag}(\mathbf{A}_{11}, \ldots, \mathbf{A}_{nn})$, we can simplify the problem to:

$$f(\lambda) = \sum_{i=1}^n \frac{y_i^2 \mathbf{A}_{ii}}{(1 + 2\lambda \mathbf{A}_{ii})^2} - 1 = 0.$$

Once $x^*$ is obtained, we can define the maximum radius of the $\ell_2$-ball centered at $b$ that does not intersect $\mathcal{R}_\mathbf{A}$ as:

$$r^* = \|(x^* + a) - b\|_2 - \epsilon,$$

for an arbitrarily small $\epsilon$. Finally, we obtain $\mathcal{R}_{\tilde{\mathbf{B}}}$ as the maximum ball contained within $\mathcal{R}_\mathbf{B}$ that has a radius smaller than $r^*$, that is:

$$\mathcal{R}_{\tilde{\mathbf{B}}} = \{x \in \mathbb{R}^n : \|x - b\|_2 \leq \min\{r^*, \min_i \mathbf{B}_{ii}\}\}.$$

Note that while choosing the radius of $\mathcal{R}_{\tilde{\mathbf{B}}}$ to be $r^*$ guarantees that $\mathcal{R}_{\tilde{\mathbf{B}}} \cap \mathcal{R}_\mathbf{A} = \emptyset$, this does not guarantee that $\mathcal{R}_{\tilde{\mathbf{B}}} \subseteq \mathcal{R}_\mathbf{B}$. To guarantee both properties, we take the minimum of both $r^*$ and $\min_i \mathbf{B}_{ii}$. This approach finds the solution to the projection of the point to the ellipsoid $\{x \in \mathbb{R}^n : x^\top \mathbf{A} x \leq 1\}$; it does not work for the case in which $b \in \mathcal{R}_\mathbf{A}$, since the problem would be trivially solved by setting $x^* = y$. Thus, our classifier must abstain in that situation.

## D.5 IMPLEMENTING LARGESTOUTSUBSET($\mathcal{R}_{\mathbf{A}}$, $\mathcal{R}_{\mathbf{B}}$) IN THE GENERALIZED CROSS-POLYTOPE CASE

Let $\mathcal{R}_{\mathbf{A}}$ and $\mathcal{R}_{\mathbf{B}}$ be two generalized cross-polytopes $\mathcal{R}_{\mathbf{A}} = \{x \in \mathbb{R}^n : \|\mathbf{A}(x - a)\|_1 \leq 1\}$ and $\mathcal{R}_{\mathbf{B}} = \{x \in \mathbb{R}^n : \|\mathbf{B}(x - b)\|_1 \leq 1\}$, where $\mathbf{A}$ and $\mathbf{B}$ are positive definite diagonal matrices with elements $\{\mathbf{A}_{ii}\}_{i=1}^n$ and $\{\mathbf{B}_{ii}\}_{i=1}^n$, respectively. The task is to find the largest possible generalized cross-polytope $\mathcal{R}_{\tilde{\mathbf{B}}}$ centered at $b$ such that $\mathcal{R}_{\tilde{\mathbf{B}}} \subseteq \mathcal{R}_{\mathbf{B}}$ where $\mathcal{R}_{\mathbf{A}} \cap \mathcal{R}_{\tilde{\mathbf{B}}} = \emptyset$.

As with the ellipsoid case, solving this problem for a generalized cross-polytope is not trivial, so instead we consider a maximum enclosing cross-polytope (i.e., $\ell_1$-ball) of $\mathcal{R}_{\tilde{\mathbf{B}}} = \{x \in \mathbb{R}^n : \|x - b\|_1 \leq r\}$ that does not intersect $\mathcal{R}_{\mathbf{A}}$ and is a subset of $\mathcal{R}_{\mathbf{B}}$. To obtain this $\ell_1$-ball, we project the center of $\mathcal{R}_{\mathbf{B}}$, $b$, to the generalized cross-polytope $\mathcal{R}_{\mathbf{A}}$ in a similar fashion to the ellipsoid case in Appendix D.4. We formulate the problem as the projection of the vector $y = b - a$ to the $\mathbf{0}_n$ centered generalized cross-polytope $\{x \in \mathbb{R}^n : \|\mathbf{A}x\|_1 \leq 1\}$.

**Lemma 4.** *Consider the hyperplane $\mathcal{H} = \{x \in \mathbb{R}^n : w^\top x - k = 0\}$ and a point $y \in \mathbb{R}^n$. The $\ell_2$ projection of $y$ on the hyperplane is the point $x^* = y - (w^\top y - k)w / \|w\|_2^2$.*

*Proof.* We define the projection problem in a similar fashion to the ellipsoid case:

$$\min_x \frac{1}{2} \|x - y\|_2^2 \qquad \text{s.t.} \quad w^\top x - k = 0,$$

and obtain the Lagrangian as $\mathcal{L}(x, \lambda) = \frac{1}{2}\|x - y\|_2^2 + \lambda(w^\top x - k)$, from where we get (using the KKT conditions): $x^* = y - \lambda^* w$ and $\lambda^* = w^\top y - k / \|w\|_2^2$; thus obtaining: $x^* = y - \frac{(w^\top y - k)w}{\|w\|_2^2}$. $\quad\square$

While this formulation does not yield the closest point from a hyperplane when measured with the $\ell_1$ norm, the fact that $\|x - x^*\|_1 \geq \|x - x^*\|_2$ implies the certification set obtained in the $\ell_1$ norm via this method is a subset of the $\ell_2$-ball of the minimum projection point. Crucially, this $\ell_2$ projection has the advantage of having a closed-form solution, while an $\ell_1$ one would require solving the problem using an iterative linear programming solver. As such, for the sake of computational complexity, we decided to use this projection, despite the sub-optimality of the result from the $\ell_1$ perspective. Empirically, we have found this does not affect our results.

Since the set of vertices of the generalized cross-polytope $\{x \in \mathbb{R}^n : \|\mathbf{A}x\|_1 \leq 1\}$ is given by $\{\mathbf{e}_i/\mathbf{A}_{ii}, -\mathbf{e}_i/\mathbf{A}_{ii}\}_{i=1}^n$, and considering the distance between the projections and the original $y$, the hyperplane that minimizes it is defined by the set of vertices $\{\text{sign}(y_i)\mathbf{e}_i/\mathbf{A}_{ii}\}_{i=1}^n$. By writing it as a system of $n$ equations, we obtain the hyperplane defined by $w = [-\text{sign}(y_1)\mathbf{A}_{11}, ..., -\text{sign}(y_n)\mathbf{A}_{nn}]$ and $k = 1$. Finally, after computing $x^*$ as per Lemma 4, we can define the maximum radius of the $\ell_1$-ball centered at $b$ that does not intersect $\mathcal{R}_{\mathbf{A}}$ as:

$$r^* = \|(x^* + a) - b\|_1 - \epsilon,$$

for an arbitrarily small $\epsilon$. Finally, and similar to the ellipsoids case, we obtain $\mathcal{R}_{\tilde{\mathbf{B}}}$ as the maximum generalized cross-polytope contained within $\mathcal{R}_{\mathbf{B}}$ that has a radius smaller than $r^*$, that is:

$$\mathcal{R}_{\tilde{\mathbf{B}}} = \{x \in \mathbb{R}^n : \|x - b\|_1 \leq \min\{r^*, \min_i \mathbf{B}_{ii}\}\}$$

Similar to before, to guarantee that the $\ell_1$ ball $\mathcal{R}_{\tilde{\mathbf{B}}}$ is still a subset to $\mathcal{R}_{\mathbf{B}}$, we take the minimum between $r^*$ and $\min_i \mathbf{B}_{ii}$ to be the radius of $\mathcal{R}_{\tilde{\mathbf{B}}}$. As with the ellipsoid case, this approach does not work for the case in which $b \in \mathcal{R}_{\mathbf{A}}$, since the assumption of the closest plane to $y$ would not hold. Thus, our classifier must abstain in that situation.

## E EXPERIMENTAL SETUP

The experiments reported in the paper used the CIFAR-10 Krizhevsky (2009)[3] and ImageNet Deng et al. (2009)[4] datasets, and trained ResNet18, WideResNet40 and ResNet50 networks He et al. (2016).

---

[3] Available here (url), under an MIT license.
[4] Available here (url), terms of access detailed in the Download page.

Experiments used the typical data split for these datasets found in the PyTorch implementation Paszke et al. (2019). The procedures to obtain the baseline networks used in the experiments are detailed in Appendix E.1 and E.2 for ellipsoids and generalized cross-polytopes, respectively. Source code to reproduce the ANCER optimization and certification results of this paper is available as supplementary material.

**Isotropic DD Optimization.** We used the available code of Alfarra et al. (2020)[5] to obtain the isotropic data dependent smoothing parameters. To train our models from scratch, we used an adapted version of the code provided in the same repository.

**Certification.** Following Cohen et al. (2019); Salman et al. (2019a); Zhai et al. (2019); Yang et al. (2020); Alfarra et al. (2020), all results were certified with $N_0 = 100$ Monte Carlo samples for selection and $N = 100,000$ estimation samples, with failure a probability of $\alpha = 0.001$.

### E.1 ELLIPSOID CERTIFICATION BASELINE NETWORKS

In terms of ellipsoid certification, the baselines we considered were COHEN Cohen et al. (2019)[6], SMOOTHADV Salman et al. (2019a)[7] and MACER Zhai et al. (2019)[8].

In the CIFAR-10 experiments, we used a ResNet18 architecture, instead of the ResNet110 used in Cohen et al. (2019); Salman et al. (2019a); Zhai et al. (2019) due to constraints at the level of computation power. As such, we had to train each of the networks from scratch following the procedures available in the source code of each of the baselines. We did so under our own framework, and the training scripts are available in the supplementary material. For the ImageNet experiments we used the ResNet50 networks provided by each of the baselines in their respective open source repositories.

We trained the ResNet18 networks for 120 epochs, with a batch size of 256 and stochastic gradient descent with a learning rate of $10^{-2}$, and momentum of 0.9.

### E.2 GENERALIZED CROSS-POLYTOPE CERTIFICATION BASELINE NETWORKS

For the certification of generalized cross-polytopes we considered RS4A Yang et al. (2020)[9]. As described in RS4A Yang et al. (2020), we take $\lambda = \sigma/\sqrt{3}$ and report results as a function of $\sigma$ for ease of comparison.

As with the baseline, we ran experiments on CIFAR-10 on a WideResNet40 architecture, and ImageNet on a ResNet50 Yang et al. (2020). However, due to limited computational power, we were not able to run experiments on the wide range of distributional parameters the original work considers, *i.e.* $\sigma = \{0.15, 0.25, 0.5, 0.75, 1.0, 1.125, 1.5, 1.75, 2.0, 2.25, 2.5, 2.75, 3.0, 3.25, 3.5\}$ on CIFAR-10 and $\sigma = \{0.25, 0.5, 0.75, 1.0, 1.125, 1.5, 1.75, 2.0, 2.25, 2.5, 2.75, 3.0, 3.25, 3.5\}$ on ImageNet. Instead, and matching the requirements from the ellipsoid section, we choose a subset of $\sigma = \{0.25, 0.5, 1.0\}$ and performed our analysis at that level.

While the trained models are available in the source code of RS4A, we ran into several issues when we attempted to use them, the most problematic of which being the fact that the clean accuracy of such models was very low in both the WideResNet40 and ResNet50 ones. To avoid these issues we trained the models from scratch, but using the stability training loss as presented in the source code of RS4A. All of these models achieved clean accuracy of over 70%.

Following the procedures described in the original work, we trained the WideResNet40 models with the stability loss used in Yang et al. (2020) for 120 epochs, with a batch size of 128 and stochastic gradient descent with a learning rate of $10^{-2}$, and momentum of 0.9, along with a step learning rate scheduler with a $\gamma$ of 0.1. For the ResNet50 networks on ImageNet, we trained them from scratch

---

[5]Data Dependent Randomized Smoothing source code available here

[6]COHEN source code available here.

[7]SMOOTHADV source code available here.

[8]MACER source code available here.

[9]RS4A source code available here.

with stability loss for 90 epochs with a learning rate of 0.1 that drops by a factor of 0.1 after each 30 epochs and a batch size of 256.

## F    SUPERSET ARGUMENT

The results we present in Section 7 support the argument that ANCER achieves, in general, a certificate that is a *superset* of the Fixed $\sigma$ and Isotropic DD ones. To confirm this at an individual test set sample level, we compare the $\ell_2$, $\ell_1$, $\ell_2^\Sigma$ and $\ell_1^\Lambda$ certification results across the different methods, and obtain the percentage of the test set in which ANCER performs at least as well as all other methods in each certificates of the samples. Results of this analysis are presented in Tables 4 and 5.

For most networks and datasets, we observe that ANCER achieves a larger $\ell_p$ certificate than the baselines in a significant portion of the dataset, showcasing the fact that it obtains a superset of the isotropic region per sample. This is further confirmed by the comparison with the anisotropic certificates, in which, for all trained networks except MACER in CIFAR-10, ANCER's certificate is superior in over 90% of the test set samples.

Table 4: Superset in top-1 $\ell_2$ and $\ell_2^\Sigma$ (rounded to nearest percent)

|  | % ANCER $\ell_2$ is the best | % ANCER $\ell_2^\Sigma$ is the best |
|---|---|---|
| CIFAR-10: COHEN | 83 | 93 |
| CIFAR-10: SMOOTHADV | 73 | 90 |
| CIFAR-10: MACER | 50 | 69 |
| ImageNet: COHEN | 94 | 96 |
| ImageNet: SMOOTHADV | 90 | 93 |

Table 5: Superset in top-1 $\ell_1$ and $\ell_1^\Lambda$ (rounded to nearest percent)

|  | % ANCER $\ell_1$ is the best | % ANCER $\ell_1^\Lambda$ is the best |
|---|---|---|
| CIFAR-10: RS4A | 100 | 100 |
| ImageNet: RS4A | 97 | 99 |

## G    EXPERIMENTAL RESULTS PER $\sigma$

### G.1    CERTIFYING ELLIPSOIDS - $\ell_2$ AND $\ell_2^\Sigma$ CERTIFICATION RESULTS PER $\sigma$

In this section we report certified accuracy at various $\ell_2$ radii and $\ell_2^\Sigma$ proxy radii, following the metrics defined in Section 7, for each training method (COHEN Cohen et al. (2019), SMOOTHADV Salman et al. (2019a) and MACER Zhai et al. (2019)), dataset (CIFAR-10 and ImageNet) and $\sigma$ ($\sigma \in \{0.12, 0.25, 0.5, 1.0\}$). Figures 6 and 7 shows certified accuracy at different $\ell_2$ radii for CIFAR-10 and ImageNet, respectively, whereas Figures 8 and 9 plot certified accuracy and different $\ell_2^\Sigma$ proxy radii for CIFAR-10 and ImageNet, respectively.

### G.2    CERTIFYING ELLIPSOIDS - $\ell_1$ AND $\ell_1^\Lambda$ CERTIFICATION RESULTS PER $\sigma$

In this section we report certified accuracy at various $\ell_1$ radii and $\ell_1^\Lambda$ proxy radii, following the metrics defined in Section 7, for RS4A, dataset (CIFAR-10 and ImageNet) and $\sigma$ ($\sigma \in \{0.25, 0.5, 1.0\}$). Figures 10 and 11 shows certified accuracy at different $\ell_1$ radii for CIFAR-10 and ImageNet, respectively, whereas Figures 12 and 13 plot certified accuracy and different $\ell_1^\Lambda$ proxy radii for CIFAR-10 and ImageNet, respectively.

### G.3    WHY DOES ANCER IMPROVE UPON ISOTROPIC DD'S $\ell_p$ CERTIFICATES?

As observed in Sections 7.1 and 7.2, ANCER's $\ell_2$ and $\ell_1$ certificates outperform the corresponding certificates obtained by Isotropic DD. To explain this, we compare the $\ell_2$ certified region obtained

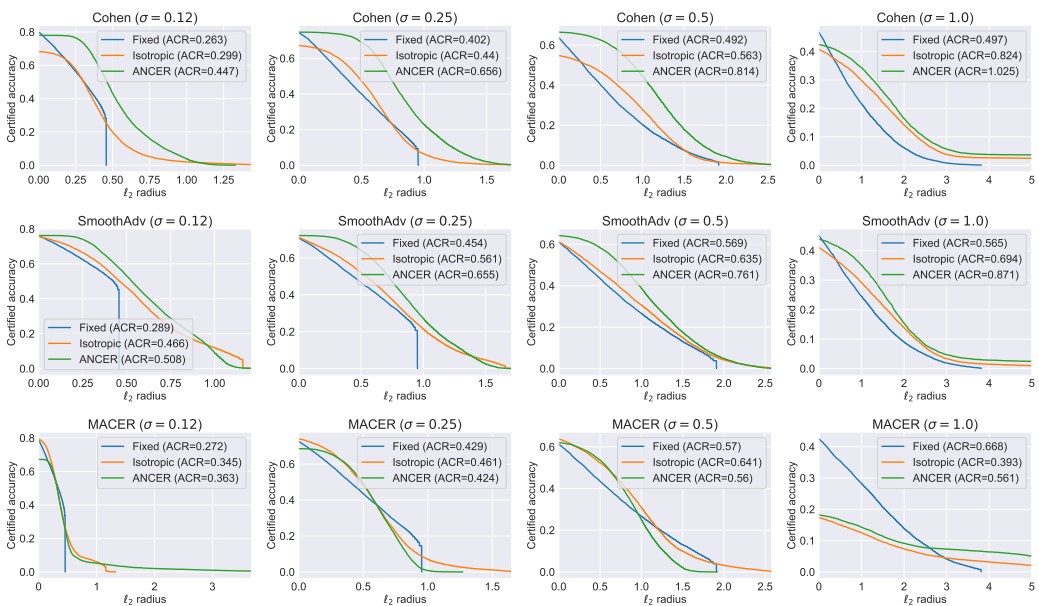

Figure 6: CIFAR-10 certified accuracy as a function of $\ell_2$ radius, per model and $\sigma$ (used as initialization in the isotropic data-dependent case and ANCER).

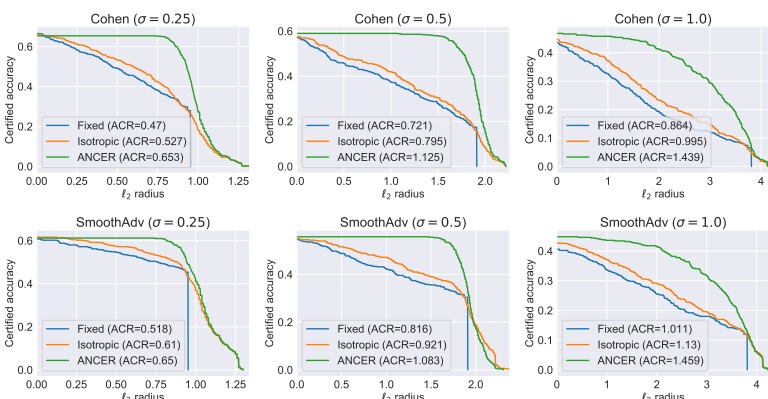

Figure 7: ImageNet certified accuracy as a function of $\ell_2$ radius, per model and $\sigma$ (used as initialization in the isotropic data-dependent case and ANCER).

by ANCER, defined in Section 6 as $\{\delta : \|\delta\|_2 \leq \min_i \sigma_i^x r(x, \Sigma^x)\}$, to the one by Isotropic DD defined as $\{\delta : \|\delta\|_2 \leq \sigma^x r(x, \sigma^x)\}$. We observe that the radius of both of these certificates can be separated into a $\sigma$-factor ($\sigma^x$ vs. $\sigma_{\min}^x = \min_i \sigma_i^x$) and a *gap*-factor ($r(x, \sigma^x)$ vs. $r(x, \Sigma^x)$). We posit the seemingly surprising result can be attributed to the computation of the gap-factor $r$ using an anisotropic, optimized distribution. However, another potential explanation would be that ANCER benefits from a prematurely stopped initialization provided by Isotropic DD, thus achieving a better $\sigma_{\min}^x$ than the isotropic $\sigma^x$ when given further optimization iterations.

To investigate this, we take the optimized parameters from the Isotropic DD experiments on SMOOTHADV for an initial $\sigma = 0.25$ on CIFAR-10, and run the optimization step of Isotropic DD for 100 iterations more than its default number of iterations from Alfarra et al. (2020), so as to match the total number of optimization steps between Isotropic DD and ANCER. The histograms of $\sigma^x$ or $\sigma_{\min}^x$ and the gap-factor $r$, *i.e.* the two factors from the $\ell_2$ certification results, are presented in Figure 14. While $\sigma^x$ for Isotropic DD is similar in distribution to ANCER's $\sigma_{\min}^x$, the distribution of the two gaps, $r(x, \sigma^x)$ and $r(x, \Sigma^x)$, are quite different.

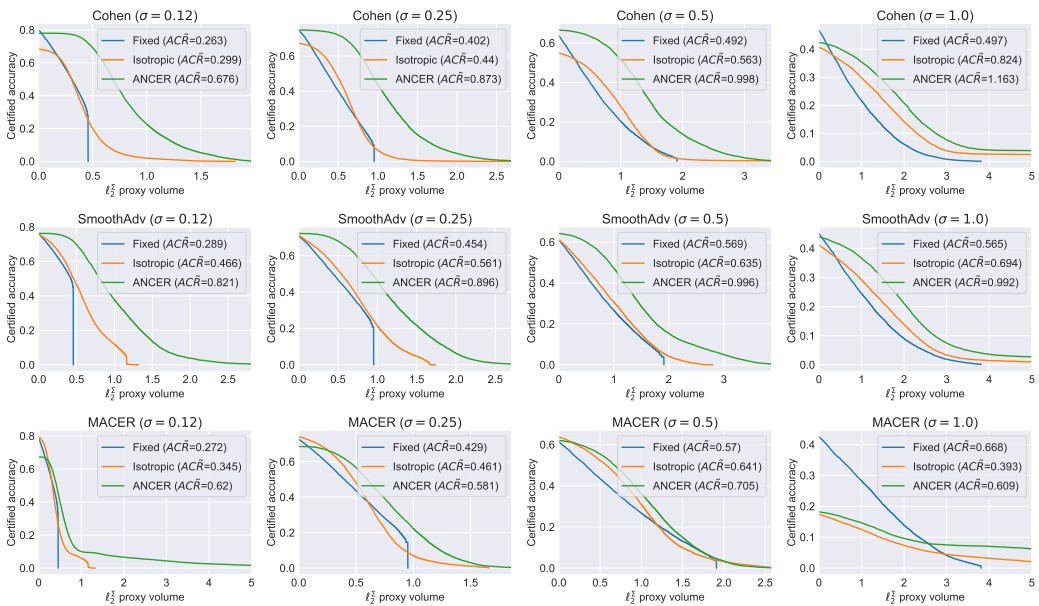

Figure 8: CIFAR-10 certified accuracy as a function of $\ell_2^{\Sigma}$ proxy radius, per model and $\sigma$ (used as initialization in the isotropic data-dependent case and ANCER).

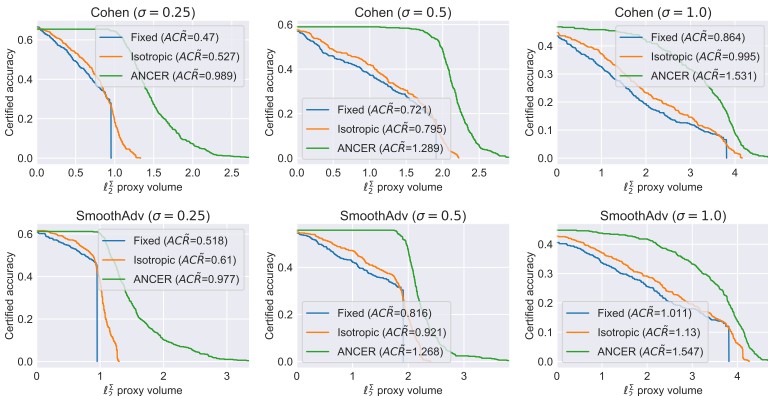

Figure 9: ImageNet certified accuracy as a function of $\ell_2^{\Sigma}$ proxy radius, per model and $\sigma$ (used as initialization in the isotropic data-dependent case and ANCER).

In particular, the ANCER certification gap is significantly larger when compared to Isotropic DD, and is the main contributor to the improvement in the $\ell_2$-ball certificate of ANCER. That is to say, ANCER generates $\Sigma^x$ that is better aligned with the decision boundaries, and hence increases the confidence of the smooth classifier.

# H    VISUAL COMPARISON OF PARAMETERS IN ELLIPSOID CERTIFICATES

Anisotropic certification allows for a better characterization of the decision boundaries of the base classifier $f$. For example, the directions aligned with the major axes of the ellipsoids $\|\delta\|_{\Sigma,2} = r$, *i.e.* locations where $\Sigma$ is large, are, by definition, expected to be less sensitive to perturbations

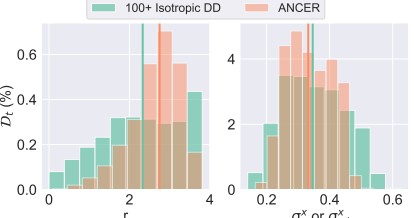

Figure 14: Histograms of the values of the $\sigma$-factor (left) and gap $r$ (right) obtained by ANCER initialized with Isotropic DD, and Isotropic DD when allowed to run for 100 iterations more than the baseline. Vertical lines plot the median of the data.

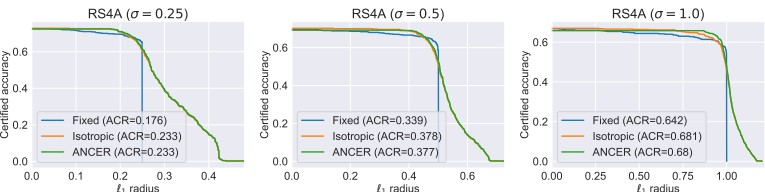

Figure 10: CIFAR-10 certified accuracy as a function of $\ell_1$ radius per $\sigma$ (used as initialization in the isotropic data-dependent case and ANCER).

Figure 11: ImageNet certified accuracy as a function of $\ell_1$ radius per $\sigma$ (used as initialization in the isotropic data-dependent case and ANCER).

compared to the minor axes directions. To visualize this concept, Figure 15 shows CIFAR-10 images along with their corresponding optimized $\ell_2$ isotropic parameters obtained by Isotropic DD, and $\ell_2^\Sigma$ anisotropic parameters obtained by ANCER. First, we note the richness of information provided by the anisotropic parameters when compared to the $\ell_2$ worst-case, isotropic one. Interestingly, pixel locations where the intensity of $\Sigma$ is large (higher intensity in Figure 15) are generally the ones corresponding least with the underlying true class and overlapping more with background pixels.

A particular insight one can get from ANCER certification is that the decision boundaries are not distributed isotropically around each input. To quantify this in higher dimensions, we plot in Figure 16 a histogram of the ratio between the maximum and minimum elements of our optimized smoothing parameters for the experiments on SmoothAdv (with an initial $\sigma = 1.0$) on CIFAR-10. We note that this ratio can be as high as 5 for some of the input points, meaning the decision boundaries in that case could be 5 times closer to a given input for some directions than others.

## I  NON DATA-DEPENDENT ANISOTROPIC CERTIFICATION

As mentioned briefly in Section 6, it is our intuition that anisotropic certification requires a data-dependent approach, as different points will have fairly different decision boundaries and the certified regions will extend in different directions (as exemplified in Figure 1).

To validate this claim, we perform certification of SmoothAdv Salman et al. (2019a) with an initial $\sigma = 1$ on CIFAR-10 using a $\Sigma$ which is the average of all the optimized $\Sigma_x$. The results of the certified accuracy, $ACR$ and $A\tilde{CR}$ are presented in Table 6, along with the same results for the methods reported in the main paper. As can be observed, moving away from the data-dependent certification in the anisotropic scenario leads to a significant performance drop in terms of robustness.

## J  THEORETICAL AND EMPIRICAL COMPARISON WITH MOHAPATRA ET AL. (2020)

In regards to the theoretical results, unfortunately the certified regions of Mohapatra et al. (2020) do not exhibit a closed form solution similarly to ours. Thus, a direct theoretical volume bound comparison is not possible.

As for the empirical comparison, ANCER's performance on both $\ell_2$ and $\ell_1$ certificates far out-does that of Mohapatra et al. (2020). For example, with $\ell_2$ certificates at a radius of $0.5$, Cohen certified with ANCER achieves $77\%$ certified accuracy (see Table 1) while Mohapatra et al. (2020) achieves under $60\%$ certified accuracy. Note that Mohapatra et al. (2020) has only a marginal improvement

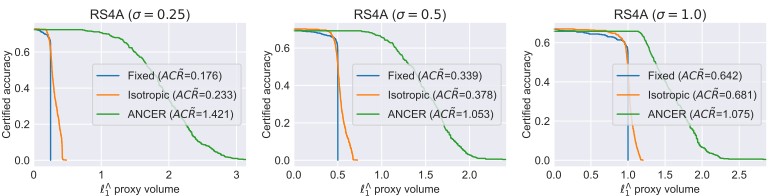

Figure 12: CIFAR-10 certified accuracy as a function of $\ell_1^\Lambda$ proxy radius per $\sigma$ (used as initialization in the isotropic data-dependent case and ANCER).

Figure 13: ImageNet certified accuracy as a function of $\ell_1^\Lambda$ proxy radius per $\sigma$ (used as initialization in the isotropic data-dependent case and ANCER).

over Cohen et al. As for the $\ell_1$ certificates, Mohapatra et al. (2020) uses the Gaussian distribution of Cohen et al, resulting in worse performance than existing state-of-art in $\ell_1$ Yang et al. (2020) that uses a uniform distribution. Our approach improves further upon the performance of Yang et al. (2020). For example, as per Table 2, RS4A with ANCER certification achieves $84\%$ certified accuracy at an $\ell_1$ radius of $0.5$, Yang et al. (2020) achieves $75\%$ certified accuracy while Mohapatra et al. (2020) achieves below $60\%$. However, we believe that the combination of both approaches, ANCER and Mohapatra et al. (2020) can further boost the performance as also hinted on in the abstract of Mohapatra et al. (2020) on the use of data-dependent smoothing.

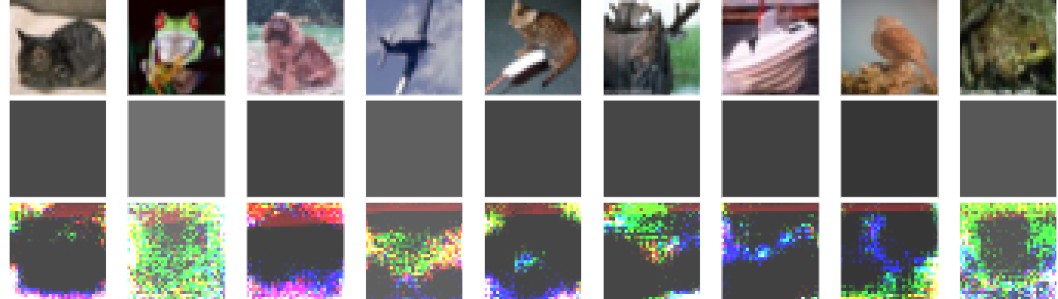

Figure 15: Visualization of an input CIFAR-10 image $x$ (top), and the optimized parameters $\sigma$ (middle) and $\Sigma$ (bottom) – higher intensity corresponds to higher $\sigma_i$ in that pixel and channel – of the smoothing distributions in the isotropic and anisotropic case, respectively.

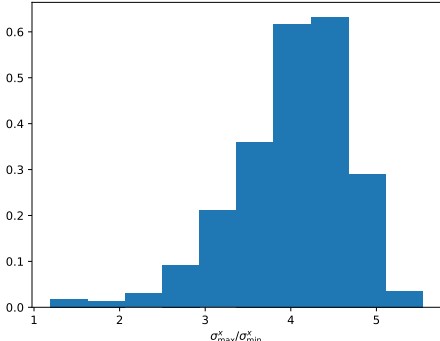

Figure 16: Distribution of the maximum over the minimum ANCER $\sigma^x$ at each dataset point for SmoothAdv Salman et al. (2019a) on CIFAR-10 (for initial $\sigma = 1.0$)

Table 6: Comparison of different certification methods on SmoothAdv with an initial $\sigma = 1.0$ on CIFAR-10.

| CIFAR-10 | SmoothAdv | Accuracy @ $\ell_2$ radius (%) | | | | | | | $\ell_2$ ACR | $\ell_2^\Sigma$ AC$\tilde{R}$ |
|---|---|---|---|---|---|---|---|---|---|---|
| | | 0.0 | 0.25 | 0.5 | 1.0 | 1.5 | 2.0 | 2.5 | | |
| | Fixed $\sigma$ | 45 | 40 | 35 | 25 | 16 | 9 | 5 | 0.565 | 0.565 |
| | Isotropic DD | 41 | 39 | 36 | 29 | 21 | 14 | 7 | 0.694 | 0.694 |
| $\sigma = 1.0$ | ANCER | 44 | 43 | 41 | 35 | 26 | 15 | 8 | **0.871** | **0.992** |
| | Average $\Sigma$ | 29 | 25 | 21 | 14 | 9 | 5 | 2 | 0.329 | 0.379 |

