# OpenReview forum: "ANCER: Anisotropic Certification  via Sample-wise Volume Maximization"
_ICLR.cc/2022/Conference — ICLR 2022 Submitted_

### Official Review · Reviewer_Fomp · 2021-10-29

**Correctness:** 4
**Technical Novelty And Significance:** 3
**Empirical Novelty And Significance:** 3
**Recommendation:** 8
**Confidence:** 4

**Main Review:**

**Strengths:**

Randomized smoothing, being the most scalable of robustness certification techniques, is an important research direction, and this work significantly advances the state of the art, even in the isotropic setting. The proposed technique is clearly explained and the analysis to justify this approach is simpler than that of Cohen or Yang. While the problem setting is slightly different than prior arts (anisotropic vs isotropic), a natural and fair quantitative comparison approach is described. The most significant and surprising result of this work: that these isotropic certificates outperform even Alfarra's approach is given ample discussion, and empirically-supported justification. The experiments are thorough and fairly compare against prior work.

**Weaknesses:**

A primary pitfall of this work is that anisotropic certificates are not particularly well-motivated. The community-accepted threat model for adversarial attackers is typically isotropic, and probing the shape of the decision boundary is of dubious utility. If this were a primary goal, more efforts would be paid to considering anisotropic certificates that are not axis-aligned (i.e. Sigma and Lambda are not diagonal). Another downside to this work is that it relies heavily upon the approach of Alfarra. Importantly, this also means AnCer inherits the main inelegance of Alfarra's approach: the memory-based classifier. This raises serious concerns in that the certificates provided are dependent upon the order in which the model is queried. While this is perhaps borderline not-kosher, it has not been a concern in practice in prior works, but it would be nice to see some evidence of this claim here.

**Questions:**

- How does Ancer perform for non axis-aligned anisotropic regions? Searching over orthogonal matrices (for the rotation) in addition to the diagonal anisotropic components could drastically increase the number of optimization variables as well as complicate the optimization landscape. How does this affect performance in terms of both runtime and reported certificates?
- Are there examples for which the memory bank of the memory-based classifier is leveraged? In particular, how does the minimum distance between a pair of cross-class test data points compare to the bounds provided?
- What insights does AnCer tell us about the shape of the decision boundary?

**Summary Of The Paper:**

The authors provide a technique for a data-dependent randomized smoothing that provides anisotropic certificates of robustness. This may be viewed as an extension of the work of Alfarra et al, where the certificates provided are axis-aligned cross-polytopes or ellipsoids; however surprisingly the proposed approach provides even tighter bounds than Alfarra's in the isotropic setting.

**Summary Of The Review:**

This approach advances the state of the art in an important line of work. A slightly nonstandard problem setting is considered, but even when comparing to the standard problem, AnCer improves upon prior works. The analysis is clean, the results surprising, and the experiments are thorough. I have no qualms recommending acceptance.

---

> ### Author Response · Authors · 2021-11-15
> **Authors' reply to reviewer Fomp**
>
> **Our Response.** We would like to thank the reviewer for the time spent carefully reviewing our paper, and for the constructive comments provided. All changes in the revised manuscript have been highlighted in blue for readability purposes.
>
> **On non axis-aligned anisotropic certification.** While we considered performing these experiments, the sheer dimensionality of the data proved to be prohibitive in terms of the runtime these would take. A possible direction we intend to pursue in the future is to consider non-axis aligned anisotropic regions for specifically designed fixed orthogonal matrix that is not learnt. For example, an orthogonal matrix that corresponds to the eigen vector space of the Hessian of the loss function at that point (similar to the analysis in Appendix A). This is motivated by that the curvature of the loss function has been shown to be intimately related to robustness (see [1]).
>
> **Minimum distance between pair of cross-class test data points.** For the memory-based post-processing to be leveraged, the overlap between certified regions must occur for differently predicted inputs. In our experiments, this only occurred for CIFAR-10 with the anisotropic $\ell_2$ certificates. Even in that case, the number of overlaps is marginal (760 samples out of 10,000), thus having a negligible effect on the overall results. Despite the distance between pairs of cross-class points here being comparable to the bounds provided, the small number of overlaps in this case is due to the optimization and certification procedures that occur before the memory-based step is applied.
>
> In the case of the anisotropic $\ell_2$ certificates for ImageNet and those of $\ell_1$ anisotropic certificates, we verified that the minimum distance between a pair of any two points is larger than the certified regions provided. For example, in ImageNet the $\ell_2$ distance between any such two points is greater than 25, while the maximum $\ell_2$ anisotropic certificate provided has a radius of less than 5.
>
> **What does ANCER tell about the shape of the decision boundaries?** ANCER's certificates allow us to understand the safe regions around each particular input point. As a corollary, it also yields the directions which are more likely to be vulnerable to an adversarial attack - visual examples of this are presented in Appendix I of the revised paper. A particular insight one can get from it is that the decision boundaries are not distributed isotropically around each input (as also highlighted in Figure 1). To quantify this in higher dimensions, we plotted a histogram of the ratio between the maximum and minimum elements of our optimized smoothing parameters for the experiments on SmoothAdv (with an initial $\sigma=1.0$) on CIFAR-10. The plot is now available in the updated manuscript in Appendix I. We have found that this ratio can be as high as 5 for some of the input points, meaning the decision boundaries in that case could be 5 times closer to a given input for some directions than others.
>
> [1] Moosavi-Dezfooli, Seyed-Mohsen, et al. "Robustness via curvature regularization, and vice versa." CVPR 2019.

---

### Official Review · Reviewer_fXE9 · 2021-11-02

**Correctness:** 3
**Technical Novelty And Significance:** 2
**Empirical Novelty And Significance:** 2
**Recommendation:** 5
**Confidence:** 4

**Main Review:**

The paper is well written and easy to follow.
At the high-level the idea of anisotropic certificates is mathematical simple, yet interesting and, as shown in the evaluation, effective.

The mathematical ideas introduced in sections 4 and 5 mostly follow directly from prior work (as is acknowledged in footnote 2).
ANCER, presented in sections 6 and C, also seems straight forward and correct. However, I have concerns about the data dependent nature of the algorithm (see below).

The evaluation section is quite thorough and obtains SOTA results.
However, what is completely missing is a discussion of the inference/prediction procedure.

Based on this my main concerns are:
- The usefulness of data-dependence approach (see below) and whether comparison with non-data-dependent approaches is fair.
- The novelty of the approach: Many of the contributions are direct consequences or applications of other approaches.
  While normally, in the light of good results, I would not mind this, in the case of this publication I am unsure, due to my first concern, how good the results really are.

Furthermore, I am wondering whether ACR gains can be realized in the anisotropic setting without per-datapoint optimization (e.g. finding a anisotropic smoothing parameters offline on the training set, and subsequently using them in certification)?

Data Dependence:
Data-dependent randomized smoothing (DDRS) [1] has been shown to be unsound in its original form and was then patched by the addition of memory-based certification in the latest draft of [1] as well as in this paper.
While the memory-based approach seems to fix the unsoundness of DDRS when applied to perturbed versions of previously certified samples, I have a few questions regarding its suitability as a practical certified defense:
- Neither this paper nor [1] discuss inference/prediction (e.g., PREDICT in [2]). Can you elaborate on how a prediction procedure would look like, that (i) is consistent with the result of the memory-based certification and (ii) is efficient (i.e. does not need to compute the certification radius at inference time)?
- Can you clarify whether the order in which inputs are presented can influence the output of the model? If so, would the model have to include a history of all samples, predictions, and certified radii previously seen? Consequently, would any obtained guarantees only be valid for a model with the exact same history, hence preventing parallel application?
- Can you discuss the possibility of an attacker deliberately presenting inputs (e.g. in small pockets of one class in the decision landscape) that influence the model’s predictions on future inputs?

Depending on these questions, data-dependent and non-data-dependent approaches (e.g. [2]) seem to target very different settings. If this is the case, I believe a direct comparison and the claim to be SOTA to be unfair.  To avoid confusion in the field, this difference in setting and all its implications should be made abundantly clear.

Further, while I am very open to discuss these points and, depending on the answers, raise my score, I am unsure whether this is the right place to discuss these issues as they mostly concern the key contributions of [1], which has not yet been published in a peer reviewed format, potentially due to these very questions.

[1] Data Dependent Randomized Smoothing, Alfarra et al.; arXiv 2020/2021

[2] Certified Adversarial Robustness via Randomized Smoothing, Cohen et al.; ICML 2019



**Summary Of The Paper:**

In this paper, the authors discuss the extension of $\ell_{p}$-randomized smoothing to anisotropic counterparts.
In particular, they consider the extension of $\ell_{2}$-certificates from (hyper)spheres to (hyper)ellipsoids
by sampling anisotropic rather than isotropic Gaussian noise, as well as the extension of $\ell_{1}$-certificates to cross-polytopes by sampling scaled Uniform noise rather than unscaled noise.
Further, the authors discuss how these extended certificates can be compared to their base-counter parts to establish superiority (inclusion).
The introduced certification algorithm, ANCER, utilizes the idea of data-dependent randomized smoothing to find the anisotropic shape with maximal certification volume.
In experimental evaluation on Cifar-10 and Imagenet the authors show that the obtained certificates permit higher isotropic certificating radii than other methods in the per-sample optimization setting.

**Summary Of The Review:**

The paper presents anisotropic robustness certificates, which seems to be technically correct and conceptually interesting.
While not stellar I believe the novelty and motivation are sufficient, given the results.
However, my reception of the results, and thereby the overall paper hinges on the data-dependent optimization procedure considered in the paper, of which I am currently not convinced.

---

> ### Author Response · Authors · 2021-11-15
> **Authors' reply to reviewer fXE9 (1/2)**
>
> **Our Response.** We would like to thank the reviewer for the time spent carefully reviewing our paper, and for the constructive comments provided. All changes in the revised manuscript have been highlighted in blue for readability purposes.
>
> **Anisotropic gains without data dependent smoothing.** We thank the reviewer for the insightful suggestion. As mentioned briefly in Section 6, we did not believe this to work very well due to the fact that different points will have fairly different nearby decision boundaries and the anisotropic certified regions will extend in different directions (as exemplified in Figure 1 of the submission).
>
> To validate this claim, we perform certification of SmoothAdv with an initial $\sigma=1$ on CIFAR-10 using a global $\Sigma$ for all inputs which is the average of all the optimized $\Sigma_x$ from the data dependent approach. The results of the certified accuracy, $ACR$ and $AC\tilde{R}$ are presented in the table below, along with the same results for the methods reported in the main paper. This comparison has also been added to the appendix of the updated manuscript.
>
> | CIFAR-10, SmoothAdv, $\sigma=1.0$ | Acc @ 0.0 (%) | Acc @ 0.25 (%) | Acc @ 0.5 (%) | Acc @ 1.0 (%) | Acc @ 1.5 (%) | Acc @ 2.0 (%) | Acc @ 2.5 (%) | $\ell_2 ACR$ | $\ell_2 AC\tilde{R}$ |
> |-----------------------------------|---------------|----------------|---------------|---------------|---------------|---------------|---------------|--------------|----------------------|
> | Fixed $\sigma$                    | 45            | 40             | 35            | 25            | 16            | 9             | 5             | 0.565        | 0.565                |
> | Isotropic DD                      | 41            | 39             | 36            | 29            | 21            | 14            | 7             | 0.694        | 0.694                |
> | ANCER                             | 44            | 43             | 41            | 35            | 26            | 15            | 8             | 0.871        | 0.992                |
> | Average $\Sigma$                  | 29            | 25             | 21            | 14            | 9             | 5             | 2             | 0.329        | 0.329                |
>
> As can be observed, removing the per-datapoint anisotropic process leads to a significant decrease in the robustness metrics.
>
> **Certification at inference time.** This is a good point. We do not have a procedure that satisfies both points requested, since inference in our case requires the certification of the point at the output prediction given for the sake of robustness. In our case, both inference and certification follow the same procedure: first certification as in [2] and then the post-certification step in Appendix D (Algorithm 1). We highlight that, while a set back, this is not a deterrent from using our method in offline scenarios where inference time is not a critical issue.

---

> > ### Author Response · Authors · 2021-11-15
> > **Authors' reply to reviewer fXE9 (2/2)**
> >
> > **Order of samples for inference and certification.** While the order in which inputs are presented in memory-based certification can influence the obtained statistics, empirically we have verified that it does not change the results significantly. However, we acknowledge this limitations and describe it along with other limitations of ANCER in the new Appendix E section of the updated manuscript.
> >
> > To properly answer the question of the effects of ordering on the output of the model, one could take all combinations of intersecting points in the test set and compute the certified accuracy, but this would be impractical given the size of the test sets considered. Instead, we first observe that the memory-based certification is order invariant in the case of no intersections of certified regions occur with samples predicted with different labels. For the case where intersections occur, we consider the "worst-case" classifier, which abstains every time an intersection of certified regions of points with different classes occurs without performing any reduction of the certified regions discussed in the Appendix. The results of such a classifier on CIFAR-10 and SmoothAdv are presented in the table below.
> >
> > | CIFAR-10, SmoothAdv | Acc @ 0.0 (%) | Acc @ 0.25 (%) | Acc @ 0.5 (%) | Acc @ 1.0 (%) | Acc @ 1.5 (%) | Acc @ 2.0 (%) | Acc @ 2.5 (%) | $\ell_2 ACR$ | $\ell_2 AC\tilde{R}$ |
> > |---------------------|---------------|----------------|---------------|---------------|---------------|---------------|---------------|--------------|----------------------|
> > | Fixed $\sigma$      | 82            | 72             | 55            | 32            | 19            | 9             | 5             | 0.834        | 0.834                |
> > | Isotropic DD        | 82            | 75             | 63            | 40            | 25            | 15            | 7             | 1.011        | 1.011                |
> > | Reported ANCER      | 83            | 81             | 73            | 48            | 30            | 17            | 8             | 1.224        | 1.573                |
> > | "Worst-case" ANCER  | 83            | 81             | 73            | 48            | 29            | 15            | 6             | 1.117        | 1.499                |
> >
> > As can be observed, even this worst-case ANCER over all possible orders outperforms both the Fixed $\sigma$ and Isotropic DD certification methods. For fairness of comparison, we plan to recompute all other results and report them in the main text of the paper instead of the random seed order previously considered (the latter ones will be moved to the appendix).
> >
> > Regarding the reviewer's second question, the memory of the model would have to include all samples, predictions and certified radii previously seen. We now discuss this in further detail in the newly added limitations section in Appendix E. While this leads to increased certification times, in our practical experiments on CIFAR-10 we observed the certification times to be manageable (Table 3).
> >
> > Finally, the only step of our method that does not lend itself to parallel computation is the post-certification step, as it requires the same history. However, we note that one could perform the expensive step of certification as in [2], and then sequentially perform the significantly cheaper post-certification step. As with the previous point, while this is a set back, it does not constitute a deterrent from using our method in offline scenarios.
> >
> > **Attacker deliberately presenting inputs in small pockets of the decision boundaries.** Given the sequential nature of the memory-based certification procedure as mentioned in the last point, such an attack would certainly be possible, provided the attacker is aware of the memory of the classifier. While we acknowledge this possibility, this attack is outside the scope of this specific work, in which we assume the adversary is only able to make changes to a single input image without knowledge of the memory of the classifier.
> >
> > However, a natural empirical defense for this type of attack would be to use a more robust base classifier. Such a classifier would be less likely to include these small pockets near the decision boundaries and thus would make such an attack harder.

---

> > > ### Comment · Reviewer_fXE9 · 2021-11-22
> > > **follow-up**
> > >
> > > I thank the authors for their reply and found especially the experiment on the averaged $\Sigma$ illustrative. I can imagine that jointly optimizing over multiple images might have produced different $\Sigma$, but can see how this is outside the scope of this publication.
> > >
> > > I applaud the authors’ efforts towards making a case for data dependence. However, while I see the points the authors are trying to make with their individual arguments, I still believe that the fundamental nature of the data-dependent approach to certification is problematic (and reviewer xsin seems to echo these concerns).
> > > My two main concerns in this regard remain:
> > > - First, I am fundamentally unsure about the implications of data-dependent certificates. The potential issues I outlined here are just a few that came to my mind and I am unsure whether further issues exist. Especially in the realm of *certified* robustness, I believe the burden of proof lies with the authors to define the exact setting in which their certificates apply and then proof that they actually do.
> > > So while I agree that arguing about the capabilities of the adversary is beyond the scope of this rebuttal, I believe it is very much necessary if the obtained radii are to be treated as certificates.
> > > - Second, the paper currently still sounds like it achieves SOTA results with no strings attached. As you point out yourself, there are several fundamental limitations (e.g., ideally an offline setting) that differentiate this approach from non-data-dependent RS. I think it would be essential to make this difference in setting abundantly clear already in the abstract, rather than burrow it in the appendix. Otherwise, these important differences might be missed by many readers.
> > >
> > > To conclude, while I find the underlying idea of anisotropic certificates highly interesting, I think that the limitations and trade-offs of the method are not made sufficiently clear to the reader. I believe that this paper’s contributions, however, would still be interesting if the setting and its limitations were openly discussed.
> > >
> > > Best,
> > > Reviewer fXE9

---

> > > > ### Author Response · Authors · 2021-11-23
> > > > **Authors' reply to reviewer fXE9**
> > > >
> > > > We thank the reviewer for their response, which we believe to be both insightful and actionable.
> > > >
> > > > We start by addressing the reviewer's main point that the limitations and trade-offs are not sufficiently clear in the paper, as well as our claim to achieving SOTA results with no strings attached. While we believe these results to be significant due to the myriad of reasons presented in our previous response, we agree that the original submission was not clear in terms of highlighting the limitations of the method (originally discussed only in appendix). As such, we decided to dedicate a subsection of Section 6 of the main paper to be on the limitations of the memory-based data-dependent certification approaches and expanded on it with feedback from all reviewers. In this new subsection, we discussed practical situations where our method is still deployable, such as the offline setup. Further, we considered the discussion about the effect of the order to our data-dependent framework, and added a comment on how we mitigated this issue in the results presented with worst-case analysis. Finally, we have also changed the paper elsewhere to qualify that the state-of-the-art results obtained are in the data-dependent setting (abstract, introduction, experiments and conclusion) so as to make the readers aware of the trade-off required to obtain the SOTA certified accuracy and ACR in the data-dependent setting.
> > > >
> > > > Regarding the reviewer's first point on the validity of the obtained certificates, we believe the description provided both in Alfarra et al. and in Appendix D of our paper points in a clear manner towards the robust-by-construction nature of these certificates in the single-image adversarial setting used in previous randomized smoothing literature. Respectfully, we disagree with the reviewer that a more exhaustive adversary description is required, given the described limitations of the method: while it opens up a new attack vector possibility, this fact does not change the soundness of our method w.r.t. the single-image adversarial setting considered.
> > > >
> > > >
> > > > We hope that the reviewer reconsiders the evaluation of our paper based on: (1) our rebuttal, (2) theoretical contributions (ease of anisotropic general analysis for arbitrary uniform and Gaussian distributions), (3) soundness of experiments that required developing several theoretical results (from reduction of ellipsoids to efficiently computing intersections) that can stand on their own, (4) certified accuracy results on several datasets, (5) extensive comparisons, (6) and the changes to the paper based on the proposed suggestions by reviewers (e.g. limitations section moved from appendix). We hope that the possibility of potential future limitations does not constitute grounds for rejection.

---

> > > > > ### Comment · Reviewer_fXE9 · 2021-11-24
> > > > > **Response**
> > > > >
> > > > > Thank you very much for your reply and the edits to the paper.
> > > > > The updated version, really addresses my key concerns as it is very open about the difference in setting.
> > > > >
> > > > > I still believe firmly that any certified defense should strive to show correctness of the method or at least state the constraints under which the method is correct, which you do now.
> > > > >
> > > > > Thus, while I won't champion the paper, I don't see ground for rejection anymore and have updated my score.

---

### Official Review · Reviewer_xsin · 2021-11-02

**Correctness:** 4
**Technical Novelty And Significance:** 3
**Empirical Novelty And Significance:** 3
**Recommendation:** 6
**Confidence:** 5

**Main Review:**

The paper is well written, polished, and easy to follow. The anistropic part of the proposed approach is well-motivated, however the sample-wise part has several issues (see below) making it unsuitable to use in practice. Moreover, some of the theoretical results are not novel (see below).

Comments:
- The derivation of the certificate in term of the Lipschitz constant under the dual norm (Proposition 1, Theorem 1) is insightful. However, this result is already known in the literature in greater generality and should be properly cited. Specifically, Theorem 1 in [1] derives the local (\alpha,\beta)-Lipschitz constant over an open set. The special case of a scalar function is also discussed (see Eq. 3 in [1]) and it coincides with the result in Proposition 1 (if we are in the special case where the open set is R). The discussion of how to use such a Lipschitz constant for certification is also discussed in [1] (see section E.2).
- The authors already discuss a know issue of data-dependent classifiers which when not tackled can lead to certificates that are not sound. To address the issue they adapt the memory-based procedure introduced in Alfarra et al. While this procedure does make the certificate sound it has other problems.
  - First, by introducing memory the proposed anisotropic certificate (or in fact any method to compute a certified region) becomes completely irrelevant. Namely, for sample i construct any region R_i and define that the prediction in that region is some arbitrary C_i. Then run the post-processing certification step in Algorithm 1 (or similarly the procedure from Alfarra et al.). Since the obtained regions will not intersect with any certified regions in the memory M the procedure is still sound. This is true regardless of how the region R_i was constructed.
  - A second, and more important, issue is that the memory makes the certificate dependent on the order of the incoming test samples. This provides a new avenue for attack, i.e. the adversary can optimize the order of the test samples to decrease the utility of the final obtained smoothed classifier.
  - Finally, the success of this memory approach also somewhat depends on the "sparsity" of the test samples. Namely, by using a small test set since the samples are in a high-dimensional space the distance between them tends to be bigger than the (proxy) radii of the certified regions. However, in a real-world application we are likely to have many more test samples which would increase the number of intersections when running Algorithm 1.
  - Note, all of these issue also apply to Alfarra et al. and are simply inherited in this approach.
- Another issue with the proposed approach is the optimization procedure described in section C. The optimization suffers from issues such as: inconsistent estimation due to clamping and not using confidence bounds, sensitivity to initialization, high gradient variance, etc. These and other issues have been described in great detail in section C.2 in [2], focusing on Alfarra et al., although, given the similarity, the same issues are inherited in this approach.
- Relatedly, in [2] the authors show that input-dependent randomized smoothing suffers from the curse of dimensionality. A discussion of this issue and how it applies to the proposed certificate should be included.
- The derivations for the special cases of certifying ellipsoids and generalized cross polytopes are useful. Note that similar results are derived in [3] (see Theorem 5 for a weighted l_2 metric equivalent to a diagonal \Sigma, and also Theorem 3) albeit in a slightly different context. A discussion on how these two relate should be included.
- The discussion of how to evaluate the anisotropic certificates and the formal definition of a "superior certificate" is appreciated.
- The experiments are executed well.

Questions:
- Assuming that estimate of the Lipschitz constant L is tight is the certificate provided by Theorem 1 also tight? For the special case of l_2 certification with "\Sigma = \sigma^2 I" tightness is shown in previous work. Does it hold in general?
- Are there any benefits to training with anisotropic noise?

References:
1. Jordan and Dimakis. "Exactly Computing the Local Lipschitz Constant of ReLU networks"
2. Sukenik et al. "Intriguing Properties of Input-dependent Randomized Smoothing"
3. Yeom and Fredrikson. "Individual Fairness Revisited: Transferring Techniques from Adversarial Robustness"

**Summary Of The Paper:**

The authors extend the standard (isotropic) l_1/l_2 randomized smoothing certificate to anisotropic smoothing distributions (and anisotropic regions). They propose to maximize the volume of the certified region for each sample independently, using a memory procedure to preserve soundness.

**Summary Of The Review:**

The paper is well written, polished, and easy to follow. The anistropic part of the proposed approach is well-motivated, however the sample-wise part has several issues (see main review) making it unsuitable to use in practice. Moreover, some of the theoretical results are not novel (see main review).

---

> ### Author Response · Authors · 2021-11-15
> **Authors' reply to reviewer xsin (1/4)**
>
> **Our Response.** We would like to thank the reviewer for the time spent carefully reviewing our paper, and for the constructive comments provided. All changes in the revised manuscript have been highlighted in blue for readability purposes.
>
> **Derivation of certificate under dual norm.** We thank the reviewer for bringing this work to our attention, as we were unaware of it. We agree that Proposition 1 and Theorem 1 are somewhat known in the literature in different contexts, particularly in the paper mentioned by the reviewer, which we have now cited in Section 4 of the revised manuscript. The earliest result of Proposition 1 we were able to find is the book [A] from 2011, on page 133 Lemma 2.6, but we believe that there is an earlier result where this is presented.
>
> We do not claim these on their own to be our theoretical contributions. At the level of the theoretical foundation, our main contribution  is on the explicit formulation of the general certification framework recipe and, as per later subsections, the ease of the specialization to the certification of ellipsoid and generalized cross-polytope. Both of which, to the best of our knowledge, are new and provide a practical solution to the problem of certification. We have updated the structure of Section 4 to reflect this point.
>
> **The irrelevance of the choice of certification regions $R_i$ in the presence of memory.** The reviewer is only partially correct here. The key metric is certified accuracy and not only the soundness of the certification. Otherwise, it would be sufficient to consider a constant classifier which enjoys an infinite radius of certification, but that does not tell us anything about accuracy.
>
> While it is the case that the memory certification is sound-by-construction, to obtain a high certified accuracy at different radii as well as high average certified radius (and additionally proxy radii in the anisotropic case), the choice of the regions $R_i$ is the key element. If one were to arbitrarily choose a certification region $R_i$ around each point, the accuracy at such small regions might be close to the clean accuracy, but the overall classifier would only enjoy a small certified accuracy. On the other hand, arbitrarily selecting large regions $R_i$ for all inputs could lead initially to robust classifiers around a handful of points that are accurate. However, as more points are certified and added to memory, the accuracy will be severely affected, dropping the overall certified accuracy. This is since many points will have their predictions incorrectly altered as they fall inside the prediction of other over estimated certified regions. Moreover, the overall certified accuracy will be affected as many other new points to be certified will have their certificates reduced due to many intersections with previous memory samples. As such, a better trade-off between robustness *vs.* accuracy is required.
>
> In our method, we propose to obtain such a balance through the anisotropic certificate derived earlier, which leads to higher accuracy at each radii than previous methods, as well as an overall higher $ACR$ and $AC\tilde{R}$.

---

> > ### Author Response · Authors · 2021-11-15
> > **Authors' reply to reviewer xsin (2/4)**
> >
> > **On the order of incoming test samples.** We agree with this observation and describe it along with other limitations of \ANCER in the new Appendix E section of the updated manuscript. While we acknowledge this possibility, this attack is outside the scope of this specific work, in which we assume the adversary is only able to make changes to a single input image without the knowledge of the memory of the classifier.
> >
> > However, the point raised by the reviewer has implications beyond the scope of an attack: how does the order affect the certified accuracy and average certified radius of the classifier under the test set? Our reported results correspond to the statistics under a random order of the test set. How much would the results be affected by different orderings? To answer this question one could take all combinations of intersecting points in the test set and compute the certified accuracy, but this would be impractical given the size of the test sets considered. Instead, we first observe that the memory-based certification is order invariant in the case of no intersections of certified regions occur with samples predicted with different labels. For the case where intersections occur, we consider the "worst-case" classifier, which abstains every time an intersection of certified regions of points with different classes occurs without performing any reduction of the certified regions discussed in the Appendix. The results of such a classifier on CIFAR-10 and SmoothAdv are presented in the table below.
> >
> > | CIFAR-10, SmoothAdv | Acc @ 0.0 (%) | Acc @ 0.25 (%) | Acc @ 0.5 (%) | Acc @ 1.0 (%) | Acc @ 1.5 (%) | Acc @ 2.0 (%) | Acc @ 2.5 (%) | $\ell_2 ACR$ | $\ell_2 AC\tilde{R}$ |
> > |---------------------|---------------|----------------|---------------|---------------|---------------|---------------|---------------|--------------|----------------------|
> > | Fixed $\sigma$      | 82            | 72             | 55            | 32            | 19            | 9             | 5             | 0.834        | 0.834                |
> > | Isotropic DD        | 82            | 75             | 63            | 40            | 25            | 15            | 7             | 1.011        | 1.011                |
> > | Reported ANCER      | 83            | 81             | 73            | 48            | 30            | 17            | 8             | 1.224        | 1.573                |
> > | "Worst-case" ANCER  | 83            | 81             | 73            | 48            | 29            | 15            | 6             | 1.117        | 1.499                |
> >
> > As can be observed, even this worst-case ANCER over all possible orders outperforms both the Fixed $\sigma$ and Isotropic DD certification methods. For fairness of comparison, we plan to recompute all other results and report them in the main text of the paper instead of the random seed order previously considered (the latter ones will be moved to the appendix).

---

> > > ### Author Response · Authors · 2021-11-15
> > > **Authors' reply to reviewer xsin (3/4)**
> > >
> > > **Sparsity of the test samples.** This statement is only partially correct. While it is true that with the increase of the number of samples the number of intersections of the memory sets would increase, it is **not** necessarily the case that the intersections between regions \textit{corresponding to different classes} will increase. As shown in Algorithm 1, set intersections are allowed as long as the corresponding regions belong to the same class.
> > >
> > > Given the anisotropic certified regions are obtained by optimizing the certified region of the smooth classifier at that specific input, the expectation is that the intersections of different predicted classes would be rare regardless of the number of test samples.
> > >
> > > We can empirically see this is the case by performing a simple experiment on the whole CIFAR-10 test set (10,000 samples). We consider the scenario of $\ell_2$ certification on SmoothAdv for $\sigma=1.0$, which, given the large starting $\sigma$, is the most likely to fall within the category of having larger proxy radii. We measure the number of times the maximum enclosing balls of the anisotropic region intersect, i.e. the function `MaxIntersect` in Algorithm 1, (a higher number suggests that the proxy radii are somewhat close to the distance between the points in $\ell_2$). We compare that against the actual number of times the $\ell_2^{\Sigma}$ ellipsoids intersect, i.e. the function `Intersect` in Algorithm 1. A high false positive rate for `MaxIntersect` would correspond to a large portion of *closer points* that have *large non-intersecting regions*. We present the results in the table below.
> > >
> > > | CIFAR-10, SmoothAdv | # `MaxIntersect` | # `Intersect` | False Positive Rate of `MaxIntersect` |
> > > |---------------------|------------------|---------------|---------------------------------------|
> > > | $\sigma=1.0$        | 5,483,082        | 760           | 99.99%                                |
> > >
> > > As can be seen, the false positive rate is close to 100\% for `MaxIntersect` given `Intersect`, meaning the inputs are actually close to each other in $\ell_2$ without intersecting in most cases. This is expected even with a growing number of points in memory due to the optimization and certification before the post-processing step.
> > >
> > > **On the issues of the optimization procedure from [2].** We have mitigated some of these issues for ANCER. This includes the clamping and improved estimation through a higher number of samples, as presented in the implementation from Listing 1 of Appendix C of our paper (improved clamping limits for efficacy) and the experimental details in Section 7 and Appendix F (we use 100 noise samples instead of 1). It is worthwhile mentioning that the small discontinuity between the exact objective to be optimized and the implementation during training is common in the literature (e.g., MACER’s official implementation deploys a similar clamping trick and the expectation is estimated with few samples as well). A similar argument for SmoothAdv with the estimation of the smooth classifier with fewer than 10 samples in some settings. Moreover, while the clamping trick might lead to linear dependence of the radius on the smoothing parameters, we implicitly clamped the smoothing parameters by using a fixed number of iterations with a small learning rate.
> > >
> > > Further, despite the validity of some of those claims, we would point to the experimental results as a validation of the empirical usefulness of the optimization as a proxy to computing high volume certified regions.
> > >
> > > **The curse of dimensionality as per [2].** We thank the reviewer for pointing out this related work. We did not include a discussion of this work in our initial submission since it appeared on arXiv after the initial deadline.
> > >
> > > Unlike [2], our methodology relies on fixed smoothing parameters within the certified region through the use of the memory. That is to say, we deployed the memory-based algorithm to make our certificate theoretically sound and, thus, the dimensionality issue raised by [2] does not directly apply to our certificate. While the problem of high dimensionality could arise in theory for input dependent smoothing for methods that do not deploy the memory-based algorithm, we found that in practice it does not materialize. The change in our results before and after deploying the memory-based algorithm is marginal. Further, the improvements on the certified accuracy that ANCER provides are far larger than the ones in the mentioned work. We will include a detailed comparison to this work including a runtime and performance comparison to the final version of our submission.

---

> > > > ### Author Response · Authors · 2021-11-15
> > > > **Authors' reply to reviewer xsin (4/4)**
> > > >
> > > > **Specialized results from [3].** We thank the reviewer for bringing this work to our attention and have modified the submission to reflect this related work. While the results presented in Theorems 5 of [3] are indeed the same as our own, the formulation and proof is significantly different and ours are simpler. In particular, the analysis in the proofs of Theorem 5 is based on Neyman-Pearson's lemma and explicitly makes use of the diagonal nature of $\Sigma$. Our analysis, on the other hand, uses the Lipschitz argument as presented at the beginning of Section 4 and our proof makes no assumptions on $\Sigma$ (other than the fact it is symmetric positive definite). Further, we point out that our framework is more general, allowing for the ease of specialization to generalized cross polytopes as well as certifying Gaussian Mixture smoothing distributions as discussed in Appendix B.1.
> > > >
> > > > **Tightness of Anisotropic certificates.** The $l_2$ anisotropic certificate is tight. This can be seen in the proof presented in Appendix B which shows the worst case base classifier $f^*$ attaining the exact upper bound for the Lipschitz constant computation.  As for the $\ell_1$ anisotropic certificate, we expect them to be tight as they coincide with RS4A in the special case of $\Lambda =\lambda I$, but we need to check.
> > > >
> > > > **Training with anisotropic noise.** Given the success of similar ideas when the training is aware of the certification procedure of previous works (e.g., Cohen, MACER and RS4A), we expect this to be the case. We plan to explore this avenue in future work.
> > > >
> > > >
> > > > [A] Shalev-Shwartz, Shai. "Online learning and online convex optimization." Foundations and trends in Machine Learning 4.2 (2011): 107-194.

---

> > > > > ### Comment · Reviewer_xsin · 2021-11-29
> > > > > **Raised Score**
> > > > >
> > > > > I thank the authors for the detailed response and the clarifications.
> > > > >
> > > > > I think future readers will benefit if some of the discussion from the response is included in the paper (e.g. in the appendix with a pointer in the main body of the paper). In particular it would be beneficial to future readers to include:
> > > > > - The discussion on the (ir)relevance of the choice of certification regions in the presence of memory
> > > > > - The discussion on (how to mitigate some of the) issues of the optimization procedure from [2]
> > > > >
> > > > > While the authors state that they "have modified the submission to reflect this related work" referring to [3], I was not able to see these changes in the latest version. Nonetheless, I trust that the authors to include these changes.
> > > > >
> > > > > Overall, taking the response and the rest of the reviews into account I have increased my score to 6. However, I strongly agree with Reviewer fXE9 that the setting and its limitations must be openly discussed. The authors already made some progress towards this goal. It would be prudent to openly include further limitations (e.g. ideally offline setting as pointed out by fXE9, and the above two discussion points) for completeness.

---

> ### Author Response · Authors · 2021-11-26
> **Follow-up with reviewer xsin**
>
> We would like once again to thank the reviewer for their original review. Given we're now fast approaching the end of the final stage of discussion, we wanted to reach out to ask if there are any questions/comments we can help address in our feedback to the original review. We would point out in particular our comment to all reviewers with the summary of changes we made to address some individual and common comments from all reviewers.

---

### Official Review · Reviewer_xapX · 2021-11-02

**Correctness:** 4
**Technical Novelty And Significance:** 3
**Empirical Novelty And Significance:** 3
**Recommendation:** 6
**Confidence:** 3

**Main Review:**

Strength

This paper is fairly interesting and insightful as it sufficiently demonstrates the advantage of introducing anisotropic robustness certificates. Although section 4.2 and 4.3, the main derivations of the non-isometric certification, are directly based on Salman et al. 2019(a), it is useful and allows less restrictive certification geometry. The discussion and adoption of Alfarra et al. 2020 in maximizing the volume through proxy radius are also interesting. Section 7.3 is especially helpful as it delves into the root keys of the better performance over isotropic DD.


Weakness

The paper mentioned all prior works considered smoothing with isotropic distributions and hence certified isotropic ell_p-ball regions, but [*] admits non-isotropic certified radius bound via first-order certification. It will be helpful if the authors can include comparisons theoretically and empirically with this prior work, allowing a more fair evaluation of the significance of the paper.

Mohapatra, Jeet, Ching-Yun Ko, Tsui-Wei Weng, Pin-Yu Chen, Sijia Liu, and Luca Daniel. "Higher-Order Certification for Randomized Smoothing." Advances in Neural Information Processing Systems 33 (2020).


Minor error: There is a redundant 'the' in the first sentence of section 7.3.

**Summary Of The Paper:**

The paper proposes the anisotropic version of randomized smoothing. Evaluation metrics based on the volume of the certified region are proposed, allowing comparisons with the certified regions provided from isotropic randomized smoothing. Experimental results show the usefulness of introducing anisotropic randomized smoothing as it certifies larger regions.

**Summary Of The Review:**

The paper contributes adequately to the community and provides enough details regarding the problem formulation and empirical performance-cost tradeoff. Including a more complete comparison with the literature will make it a stronger submission.

---

> ### Author Response · Authors · 2021-11-15
> **Authors' reply to reviewer xapX**
>
> **Our Response.** We would like to thank the reviewer for the time spent carefully reviewing our paper, and for the constructive comments provided. All changes in the revised manuscript have been highlighted in blue for readability purposes.
>
> **Theoretical and empirical comparisons to [\*].** In regards to the theoretical results, unfortunately the certified regions of [\*] do not exhibit a closed form solution similarly to ours. Thus, a direct theoretical volume bound comparison is not possible.
>
> As for the empirical comparisons, our performance on both $\ell_2$ and $\ell_1$ certificates far out-does that of [\*]. For example, with $\ell_2$ certificates at a radius of $0.5$, Cohen certified with ANCER achieves $77\%$ certified accuracy (see Table 1) while [\*] achieves under $60\%$ certified accuracy. Note that [\*] has only a marginal improvement over Cohen et al. As for the $\ell_1$ certificates, [\*] uses the Gaussian distribution of Cohen et al, resulting in worse performance than existing state-of-art in $\ell_1$ [A] that uses uniform distribution. Our approach improves further upon the performance of [A]. For example, as per Table 2, RS4A with ANCER certification achieves $84\%$ certified accuracy at an $\ell_1$ radius of $0.5$, [A] achieves $75\%$ certified accuracy while [*] achieves below $60\%$. However, we believe that the combination of both approaches, ANCER and [\*] can further boost the performance as also hinted on in the abstract of [\*] on the use of data-dependent smoothing.
>
> Given the relevance of the comparison, we have updated our submission to include a reference to this paper in Section 2, as well as added this discussion of the comparison with [\*] in Appendix K.
>
>
>
> [A] "Randomized smoothing of all shapes and sizes." Greg Yang, Tony Duan, J Edward Hu, Hadi Salman, Ilya Razenshteyn, and Jerry Li.  ICML20.
>
> [*] "Higher-Order Certification for Randomized Smoothing." Mohapatra, Jeet, Ching-Yun Ko, Tsui-Wei Weng, Pin-Yu Chen, Sijia Liu, and Luca Daniel.  NeurIPS20.

---

### Author Response · Authors · 2021-11-23
**Summary of Changes**

We would like to thank all the reviewers for their comments and discussion so far. We believe the provided feedback has improved our submission significantly. In our response, we did our best to address all concerns by providing several experiments validating our claims. The summary of changes follows.


- We have moved the discussion of the limitations of the method and settings from the appendix to Section 6 of the main paper. This section is a result of the discussion with some of the reviewers. It showcases some of the setbacks of the memory-based data-dependent setting we operate in, as well as the use cases and mitigation strategies which make it a relevant solution.

- We have qualified our state-of-the-art results in several sections of the paper (abstract, introduction, experiments and conclusion) to take into account the trade-off between the improvements in certified accuracy and radii, and the limitations described in Section 6.


- We have modified the theoretical introduction in Section 4 to highlight our particular claims and improve the separation from the previously known results.

- We moved the discussion of why ANCER outperforms Isotropic DD in the $\ell_2$ scenario from Section 7.3 to Appendix G.3 to comply with the 9-pages limit.

- We have added Appendix I as a study of non data-dependent anistropic certification (experiments suggested by fXE9), as well as a theoretical and empirical comparison to Mohapatra et al. (2020) (suggested by xapX) in Appendix J.


Please let us know if there are any other questions or experiments that you think are necessary to improve the review scores. We look forward to continuing the discussion over the rest of the rebuttal period.

---

### Decision · Program_Chairs · 2022-01-20

**Decision:**

Reject

**Comment:**

The paper proposes the anisotropic version of randomized smoothing. Evaluation metrics based on the volume of the certified region are proposed, allowing comparisons with the certified regions provided from isotropic randomized smoothing. Experimental results show the usefulness of introducing anisotropic randomized smoothing as it certifies larger regions.

Strengths:
+ The paper is well written, polished, and easy to follow.
+ The anisotropic part of the proposed approach is well-motivated.
+ The evaluation section is quite thorough and obtains SOTA results.

Weaknesses:
- The sample-wise (data dependent) part has several issues making it unsuitable to use in practice. The authors already discuss a know issue of data-dependent classifiers which when not tackled can lead to certificates that are not sound. To address the issue they adapt the memory-based procedure introduced in Alfarra et al. While this procedure does make the certificate sound it has other problems. For example, an issue is that the memory makes the certificate dependent on the order of the incoming test samples. This provides a new avenue for attack, i.e. the adversary can optimize the order of the test samples to decrease the utility of the final obtained smoothed classifier. In addition, the success of this memory approach also somewhat depends on the "sparsity" of the test samples. Namely, by using a small test set since the samples are in a high-dimensional space the distance between them tends to be bigger than the (proxy) radii of the certified regions. However, in a real-world application we are likely to have many more test samples which would increase the number of intersections when running Algorithm 1. Although the authors provide opposite empirical evidence on a specific dataset, it is not very clear how general it is for other datasets.

- Some of the theoretical results are not novel as they follow directly from prior work (as acknowledged in the paper).

- Another issue with the proposed approach is the optimization procedure described in section C. The optimization suffers from issues such as: inconsistent estimation due to clamping and not using confidence bounds, sensitivity to initialization, high gradient variance, etc.

---

> ### Public Comment · ~Francisco_Eiras1 · 2022-02-08
> **Clarifications on the PC's mentioned weaknesses**
>
> We appreciate the time spent by the PC to comment on the decision taken. However, we would like to clarify some points regarding the weaknesses pointed out by the PC:
>
> Regarding the PC's first comments on the memory-based aspect of the classifier introduced in Alfarra et al., we respectfully disagree with the PC that the classifier is not usable in practice. We acknowledge the fact that the resulting classifier can be dependent on the order of the incoming test samples in the main text of the paper. Empirically, we have verified that this is not a significant issue, and changed the reporting of results to a "worst-case" analysis (i.e. assuming the order which leads to the worst performance), noting our method still beats the baselines in this setting. Further, as noted in the rebuttal, we do not believe the success of the approach is necessarily dependent on the size of the test set. Given the ANCER optimization, the number of intersections occurring in Algorithm 1 should mostly occur for regions classified with the same label even in dense spaces - as shown in the experiment done for the author response to reviewer xsin's original comment - which does not negatively affect the certified accuracy obtained by the method.
>
> Regarding the PC's comment on the novelty of some of the theoretical results, we believe this to be a moot case. It is natural that a new work should base its results on previously existing theory. As the PC mentioned, we do not claim these results to be our own contribution. Instead, we claim that the connection between the Lipschitz constant and the robustness certificate is the base on which we build the derivation for the new results that follow (the certification of ellipsoids and generalized cross-polytopes).
>
> Regarding the PC's last comment on the issue with the optimization procedure, we believe some of these issues simply are not verified in practice. A small discontinuity between the exact objective to be optimized and the implementation during training is common in the literature (e.g., MACER's official implementation deploys a similar clamping trick and the expectation is estimated with few samples as well). A similar argument is present for SmoothAdv with the estimation of the smooth classifier with fewer than 10 samples in some settings. Moreover, while the clamping trick might lead to linear dependence of the radius on the smoothing parameters, we implicitly clamped the smoothing parameters by using a fixed number of iterations with a small learning rate. As such, despite the validity of some of those claims, we would point to the experimental results as a validation of the empirical usefulness of the optimization as a proxy to computing high volume certified regions.
>
> During the rebuttal process, we were able to answer all of the reviewers' questions, either by providing solid justifications or via satisfactory empirical evidence. Through this, we believe the reviewers were convinced with the arguments presented - as supported by two score changes (one 3 to 5 and the other 5 to 6) and final scores of the paper.